# SLOWMO: IMPROVING COMMUNICATION-EFFICIENT DISTRIBUTED SGD WITH SLOW MOMENTUM

**Jianyu Wang**[*]
Department of Electrical and Computer Engineering
Carnegie Mellon University
Pittsburgh, PA 15213, USA
`jianyuw1@andrew.cmu.edu`

**Vinayak Tantia, Nicolas Ballas & Michael Rabbat**
Facebook AI Research
Montreal, Canada
`{tantia, ballasn, mikerabbat}@fb.com`

## ABSTRACT

Distributed optimization is essential for training large models on large datasets. Multiple approaches have been proposed to reduce the communication overhead in distributed training, such as synchronizing only after performing multiple local SGD steps, and decentralized methods (*e.g.*, using gossip algorithms) to decouple communications among workers. Although these methods run faster than ALLREDUCE-based methods, which use blocking communication before every update, the resulting models may be less accurate after the same number of updates. Inspired by the BMUF method of Chen & Huo (2016), we propose a *slow momentum* (SLOWMO) framework, where workers periodically synchronize and perform a momentum update, after multiple iterations of a base optimization algorithm. Experiments on image classification and machine translation tasks demonstrate that SLOWMO consistently yields improvements in optimization and generalization performance relative to the base optimizer, even when the additional overhead is amortized over many updates so that the SLOWMO runtime is on par with that of the base optimizer. We provide theoretical convergence guarantees showing that SLOWMO converges to a stationary point of smooth non-convex losses. Since BMUF can be expressed through the SLOWMO framework, our results also correspond to the first theoretical convergence guarantees for BMUF.

## 1 INTRODUCTION

Distributed optimization (Chen et al., 2016; Goyal et al., 2017) is essential for training large models on large datasets (Radford et al., 2019; Liu et al., 2019; Mahajan et al., 2018b). Currently, the most widely-used approaches have workers compute small mini-batch gradients locally, in parallel, and then aggregate these using a blocking communication primitive, ALLREDUCE, before taking an optimizer step. Communication overhead is a major issue limiting the scaling of this approach, since ALLREDUCE must complete before every step and blocking communications are sensitive to stragglers (Dutta et al., 2018; Ferdinand et al., 2019).

Multiple complementary approaches have recently been investigated to reduce or hide communication overhead. Decentralized training (Jiang et al., 2017; Lian et al., 2017; 2018; Assran et al., 2019) reduces idling due to blocking and stragglers by employing approximate gradient aggregation (*e.g.*, via gossip or distributed averaging). Approaches such as Local SGD reduce the frequency of communication by having workers perform multiple updates between each round of communication (McDonald et al., 2010; McMahan et al., 2017; Zhou & Cong, 2018; Stich, 2019; Yu et al., 2019b). It is also possible to combine decentralized algorithms with Local SGD (Wang & Joshi,

---

[*]Work performed while doing an internship at Facebook AI Research.

2018; Wang et al., 2019). These approaches reduce communication overhead while injecting additional noise into the optimization process. Consequently, although they run faster than large mini-batch methods, the resulting models may not achieve the same quality in terms of training loss or generalization accuracy after the same number of iterations.

Momentum is believed to be a critical component for training deep networks, and it has been empirically demonstrated to improve both optimization and generalization (Sutskever et al., 2013). Yet, there is no consensus on how to combine momentum with communication efficient training algorithms. Momentum is typically incorporated into such approaches by having workers maintain separate buffers which are not synchronized (Lian et al., 2017; 2018; Assran et al., 2019; Koloskova et al., 2019a). However, recent work shows that synchronizing the momentum buffer, using periodic ALLREDUCE or a decentralized method, leads to improvements in accuracy at the cost of doubling the communication overhead (Yu et al., 2019a). In *block-wise model update filtering* (BMUF), nodes perform multiple local optimization steps between communication rounds (similar to local SGD), and they also maintain a momentum buffer that is only updated after each communication round (Chen & Huo, 2016). Although it is now commonly used for training speech models, there are no theoretical convergence guarantees for BMUF, and it has not been widely applied to other tasks (*e.g.*, in computer vision or natural language processing).

Inspired by BMUF, we propose a general framework called *slow momentum* (SLOWMO) to improve the accuracy of communication-efficient distributed training methods. SLOWMO runs on top of a base algorithm, which could be local SGD or a decentralized method such as *stochastic gradient push* (SGP) (Nedić & Olshevsky, 2016; Assran et al., 2019). Periodically, after taking some number $\tau$ of base algorithm steps, workers average their parameters using ALLREDUCE and perform a momentum update. We demonstrate empirically that SLOWMO consistently improves optimization and generalization performance across a variety of base algorithms on image classification and neural machine translation tasks—training ResNets on CIFAR-10 and ImageNet, and training a transformer on WMT'16 En-De. Ultimately, SLOWMO allows us to reap the speedup and scaling performance of communication-efficient distributed methods without sacrificing as much in accuracy.

We also prove theoretical bounds showing that SLOWMO converges to a stationary point of smooth non-convex functions at a rate $\mathcal{O}(1/\sqrt{mT\tau})$ after $T\tau$ total inner optimization steps and $T$ SLOWMO updates with $m$ worker nodes, for a variety of base optimizers. Thus, SLOWMO is order-wise no slower than stochastic gradient descent. BMUF and the recently-proposed Lookahead optimizer (Zhang et al., 2019) can be expressed through the SLOWMO framework, and so our results also translate to the first theoretical convergence guarantees for both of these methods.

## 2 THE SLOW MOMENTUM (SLOWMO) FRAMEWORK

SLOWMO is a framework intended for solving stochastic optimization problems of the form

$$\min_{\boldsymbol{x} \in \mathbb{R}^d} \frac{1}{m} \sum_{i=1}^{m} \mathbb{E}_{\xi_i \sim D_i} F_i(\boldsymbol{x}; \xi_i), \tag{1}$$

using $m$ worker nodes, where the loss function term $F_i$ and samples $\xi_i$ from the distribution $D_i$ are available at the $i$th worker. SLOWMO builds on top of a base optimization algorithm and has a nested loop structure shown in Algorithm 1. Each worker maintains a local copy of the parameters, $\boldsymbol{x}_{t,k}^{(i)}$ at worker $i$ after the $k$th inner step of the $t$th outer iteration. We assume that all workers are initialized to the same point $\boldsymbol{x}_{0,0}$, and the framework also uses a slow momentum buffer $\boldsymbol{u}_t$ which is initialized to $\boldsymbol{u}_0 = \boldsymbol{0}$; although each worker stores a copy of $\boldsymbol{u}_t$ locally, these are always synchronized across all nodes, so we omit the superscript to simplify the notation.

Within each outer iteration, workers first take $\tau$ steps of the base optimizer. The base optimizer could be a method which involves no communication, such as SGD (with or without momentum) or a decentralized algorithm which involves some communication, such as *stochastic gradient push* (SGP) (Assran et al., 2019). We denote these updates by $\boldsymbol{x}_{t,k+1}^{(i)} = \boldsymbol{x}_{t,k}^{(i)} - \gamma_t \boldsymbol{d}_{t,k}^{(i)}$ where $\gamma_t$ is the base optimizer (fast) learning rate and $\boldsymbol{d}_{t,k}^{(i)}$ is the update direction used at worker $i$. If the base optimizer is SGD then $\boldsymbol{d}_{t,k}^{(i)} = \nabla F_i(\boldsymbol{x}_{t,k}^{(i)}; \xi_{t,k}^{(i)})$. For other base optimizers which may use additional

---

**Algorithm 1:** Slow Momentum

---

**Input:** Base optimizer with learning rate $\gamma_t$; Inner loop steps $\tau$;
      Slow learning rate $\alpha$; Slow momentum factor $\beta$;
      Number of worker nodes $m$. Initial point $\boldsymbol{x}_{0,0}$ and
      initial slow momentum buffer $\boldsymbol{u}_0 = \boldsymbol{0}$.

1 **for** $t \in \{0, 1, \ldots, T-1\}$ **at worker** $i$ **in parallel do**
2     Reset/maintain/average base optimizer buffers
3     **for** $k \in \{0, 1, \ldots, \tau-1\}$ **do**
4        Base optimizer step: $\boldsymbol{x}_{t,k+1}^{(i)} = \boldsymbol{x}_{t,k}^{(i)} - \gamma_t \boldsymbol{d}_{t,k}^{(i)}$
5     **end**
6     Exact-Average: $\boldsymbol{x}_{t,\tau} = \frac{1}{m} \sum_{i=1}^{m} \boldsymbol{x}_{t,\tau}^{(i)}$
7     Update slow momentum: $\boldsymbol{u}_{t+1} = \beta \boldsymbol{u}_t + \frac{1}{\gamma_t} \left( \boldsymbol{x}_{t,0} - \boldsymbol{x}_{t,\tau} \right)$
8     Update outer iterates: $\boldsymbol{x}_{t+1,0} = \boldsymbol{x}_{t,0} - \alpha \gamma_t \boldsymbol{u}_{t+1}$
9 **end**

---

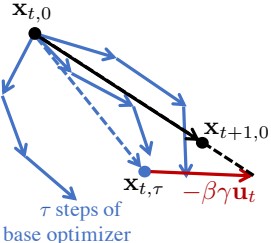

Figure 1: Illustration of one outer iteration in the slow momentum framework for $m = 3$ workers.

local momentum or communication, $\boldsymbol{d}_{t,k}^{(i)}$ represents the full update applied at worker $i$ on this step. Specific examples of $\boldsymbol{d}_{t,k}^{(i)}$ for different base optimizers are presented in Table C.1 in Appendix C.

After the $\tau$ base optimizer steps, the workers calculate the average $\boldsymbol{x}_{t,\tau} = \boldsymbol{x}_{t,0} - \frac{\gamma_t}{m} \sum_{i=1}^{m} \sum_{k=0}^{\tau-1} \boldsymbol{d}_{t,k}^{(i)}$ using ALLREDUCE (line 6), and then they perform a slow momentum update (lines 7–8),

$$\boldsymbol{u}_{t+1} = \beta \boldsymbol{u}_t + \frac{1}{\gamma_t} \left( \boldsymbol{x}_{t,0} - \boldsymbol{x}_{t,\tau} \right) \tag{2}$$

$$\boldsymbol{x}_{t+1,0} = \boldsymbol{x}_{t,0} - \alpha \gamma_t \boldsymbol{u}_{t+1}. \tag{3}$$

Although the workers perform this update locally, in parallel, we again omit superscripts because the values of $\boldsymbol{x}_{t,0}$, $\boldsymbol{x}_{t,\tau}$, and hence $\boldsymbol{u}_{t+1}$ and $\boldsymbol{x}_{t+1,0}$ are always identical across all workers, since they follow the ALLREDUCE in line 6. Note that the difference $\boldsymbol{x}_{t,0} - \boldsymbol{x}_{t,\tau}$ is scaled by $\frac{1}{\gamma_t}$ in (2) to make the slow momentum buffer invariant to the fast learning rate $\gamma_t$, which may change through training, *e.g.*, when using a learning rate schedule. The outer update in line 8 uses the product $\alpha \gamma_t$ of the slow and fast learning rates. We use the distinction between slow and fast because the base optimizer step is applied $\tau$ times for each outer update, but this is not intended to imply that one learning rate is necessarily bigger or smaller than the other. We give specific examples of learning rates and other hyperparameters used in the experiments in Section 4 below.

A specific SLOWMO algorithm instance is obtained by specifying the base algorithm and the hyperparameters $\alpha$, $\beta$, $\gamma$, and $\tau$. We can recover a number of existing algorithms in this framework. When the base algorithm is SGD, $\tau = 1$, $\alpha = 1$, and $\beta \in [0, 1)$, we recover standard large mini-batch SGD with learning rate $\gamma$ and momentum $\beta$. When the base algorithm is SGD, $\tau > 1$, $\alpha = 1$, and $\beta = 0$, we recover Local SGD (McDonald et al., 2010; Stich, 2019; Yu et al., 2019b; Wang & Joshi, 2018). When the base algorithm does not involve communication among workers, $\tau > 1$ and $\beta > 0$, we recover BMUF (Chen & Huo, 2016).

We also obtain interesting novel distributed algorithms. In particular, the experiments in Section 4 demonstrate that using SLOWMO with a decentralized base algorithm like SGP and reasonable values of $\tau$ consistently leads to improved optimization and generalization performance over the base method alone, without a significant increase in runtime. We also observe empirically that, for a fixed number of iterations, SLOWMO combined with SGP is superior to SLOWMO combined with SGD.

The above are all distributed algorithms. Perhaps surprisingly, SLOWMO also encompasses a recently-introduced non-distributed method: if we have $m = 1$ worker with SGD/Adam as the base algorithm, $\alpha \in (0, 1]$, $\beta = 0$, and $\tau > 0$, we recover the Lookahead optimizer of Zhang et al. (2019), which also has a nested loop structure. Section 5 provides theoretical convergence guarantees when using the SLOWMO framework to minimize smooth non-convex functions, and thus provides the first theoretical convergence guarantees in the literature for BMUF and Lookahead in this setting.

## 3 RELATED WORK

The idea of reducing communication overhead by using ALLREDUCE to synchronize parameters after every $\tau > 0$ optimizer steps has been considered at least since the work of McDonald et al. (2010), and has been more recently referred to as Local SGD in the literature. Elastic-average SGD (Zhang et al., 2015) uses a related approach, but with a parameter server rather than ALLREDUCE. Lin et al. (2018) apply Local SGD for distributed training of deep neural networks and propose post-local SGD, which starts by running ALLREDUCE-SGD for some epochs before switching to Local SGD, to improve generalization at the cost of additional communication.

Decentralized methods use approximate distributed averaging over a peer-to-peer topology, rather than ALLREDUCE. This decouples communication but also injects additional noise in the optimization process since the models at different workers are no longer precisely synchronized. Lian et al. (2017) present *decentralized parallel SGD* (D-PSGD), where each worker sends a copy of its model to its peers at every iteration, and show it can be faster than parameter-server and ALLREDUCE methods for training deep neural networks. Lian et al. (2018) study an *asynchronous* extension, AD-PSGD. Assran et al. (2019) study *stochastic gradient push* (SGP), and propose its asynchronous counterpart *overlap SGP* (OSGP), which achieve a further speedup over D-PSGD and AD-PSGD by using less coupled communication. D-PSGD, AD-PSGD, and SGP all have similar theoretical convergence guarantees for smooth non-convex functions, showing a *linear scaling* relationship between the number of workers and the number of iterations to reach a neighborhood of a first-order stationary point. Although the theory for all three methods only covers the case of SGD updates without momentum, implementations use momentum locally at each worker, and workers only average their model parameters (not momentum buffers). Yu et al. (2019a) prove that linear scaling holds when workers average their parameters *and* momentum buffers, although this doubles the communication overhead. We refer to this approach as *double-averaging* below.

Scaman et al. (2019) establish optimal rates of convergence for decentralized optimization methods in the deterministic, convex setting. Richards & Rebeschini (2019) provide guarantees on the generalization error of non-parametric least-squares regression trained using decentralized gradient descent, showing that there are regimes where one can achieve a linear speedup. Neither of these results apply directly to the setting considered in this paper, which focuses on smooth non-convex stochastic optimization, and extending this line of work to non-convex settings is an interesting direction.

Mahajan et al. (2018a) propose an approach to distributed learning of linear classifiers (*i.e.*, convex problems) where, in parallel, workers minimize locally formed approximate loss functions, and then the resulting minimizers are averaged to determine a descent direction. Methods which fit in the SLOWMO framework, including Local SGD, BMUF (Chen & Huo, 2016), and the serial Lookahead optimizer (Zhang et al., 2019), can be seen as related to this approach, where the actual loss function at each worker is used rather than an approximate one, and where the descent direction is used in a momentum update rather than a (deterministic) line search method.

Note that various approaches to gradient compression have been proposed to reduce the communication overhead for ALLREDUCE and decentralized learning methods (Alistarh et al., 2007; Wen et al., 2007; Bernstein et al., 2019; Karimireddy et al., 2019; Koloskova et al., 2019b; Vogels et al., 2019). However, it is presently not clear to what extent compression may be beneficial for methods like BMUF, D-PSGD, SGP, and OSGP, which perform averaging on the model parameters rather than on gradients. Combining SLOWMO with compression techniques is an interesting and important direction for future work.

Finally, although momentum methods are known to achieve accelerated rates for deterministic optimization, currently the theoretical understanding of the benefits of momentum methods (both serial and parallel) is limited (Bottou et al., 2018). Loizou & Richtárik (2017) show that accelerated convergence rates can be achieved when the objective is a quadratic finite sum (i.e., least squares problem) that satisfies an interpolation condition. Can et al. (2019) show that accelerated rates can also be achieved in the more general setting of smooth, strongly convex objectives, in a regime where the noise level is below an explicit threshold, and where the rate of convergence is measured in terms of the 1-Wasserstein metric of the distribution of trajectories produced by the momentum method. In the setting of smooth non-convex functions, Gitman et al. (2019) establish stability and asymptotic convergence results for the quasi-hyperbolic momentum method (Ma & Yarats, 2019),

which can be viewed as interpolating between SGD and a stochastic momentum method. Extending these results, which focus on the serial setting, to obtain decentralized momentum methods with guaranteed acceleration, is an important direction for future work.

## 4 EXPERIMENTAL RESULTS

We evaluate the effectiveness of SLOWMO on three datasets: image classification on CIFAR-10 and ImageNet, and neural machine translation on WMT'16-En-De. All experiments use NVIDIA DGX-1 servers as worker nodes. Each server contains 8 NVIDIA V100 GPUs and the servers are internetworked via commodity 10 Gbps Ethernet.

On CIFAR-10 (Krizhevsky et al., 2009), we train a ResNet-18 (He et al., 2016) using 32 V100 GPUs, located on 32 different worker nodes. The total mini-batch size is 4096, and we train for 200 epochs. The learning rate ($\gamma_t$) linearly increases during the first 5 epochs, following the warm-up strategy in Goyal et al. (2017), and then decays by a factor of 10 at epochs 100, 150, and 175. The (fast) learning rate was tuned separately for each base optimizer. All experiments were run 5 times with different random seeds, and the mean metrics are reported.

On ImageNet (Krizhevsky et al., 2012), we train a ResNet-50 (He et al., 2016) using 32 worker nodes (*i.e.*, 256 GPUs). The total mini-batch size is 8192, and we train for 90 epochs. The learning rate schedule is identical to (Goyal et al., 2017), *i.e.*, linear warm-up in the first 5 epochs and decay by a factor of 10 at epochs 30, 60 and 80.

On WMT'16-En-De, we train a transformer model (Vaswani et al., 2017) using 8 worker nodes (*i.e.*, 64 GPUs). The model is trained with 200k token batches, and we train for 25 epochs. We follow the experimental setting of Ott et al. (2018).

For each task, we consider several baselines: (i) Local SGD /Local Adam, where worker nodes independently run single-node SGD (with Nesterov momentum) or Adam and periodically average model parameters; (ii) stochastic gradient push (SGP), the state-of-the-art synchronous decentralized training method; and (iii) Overlap-SGP (OSGP), an asynchronous version of SGP. For each baseline, we examine its performance with and without SLOWMO. Recall that Local SGD and Local Adam with SLOWMO are equivalent to BMUF. Local SGD and Local Adam do not involve communication during the inner loop (base optimizer) updates, while SGP and OSGP involve gossiping with one peer at every step. In addition, we also evaluate the performance of AR-SGD/AR-Adam, the traditional ALLREDUCE implementation of parallel SGD/Adam. Details of all baseline methods are provided in Appendices A and C.

In general, the hyperparameters of SLOWMO (slow learning rate $\alpha$, slow momentum $\beta$, and number of inner loop steps $\tau$) need to be tuned for each base optimizer and task. The results in Table 1 all use $\alpha = 1$, which we found to be consistently the best. For Local SGD (with or without SLOWMO), we set $\tau = 12$, and for all other baseline methods we use $\tau = 48$. Using $\tau > 12$ for Local SGD resulted in significantly worse loss/accuracy on ImageNet and WMT'16 En-De.

Note also that all of our baselines (or base optimizers) leverage a local momentum scheme, following previous works (Assran et al., 2019; Koloskova et al., 2019a). When using these methods with SLOWMO, there are different ways to handle the base algorithm local momentum buffers at the beginning of each outer loop (line 2 in Algorithm 1): zeroing, averaging among workers, or maintaining the current local value. Appendix B.4 provides an empirical comparison. For the experiments reported here, when using SGD with Nesterov momentum as the base algorithm (CIFAR-10 and Imagenet) we zero the base algorithm buffer, and when using Adam as the base algorithm (WMT'16 En-De) we maintain the current value of the Adam buffers. We also tried to apply SLOWMO on top of AR-SGD base optimizer, but we did not observe any improvement in that setting.

**Optimization and Generalization Performance.** Table 1 shows the best training loss and the validation accuracy/BLEU score for each baseline, with and without SLOWMO. Using SLOWMO consistently improves both the optimization and generalization performance across all training tasks and baseline algorithms. Figure 2 presents validation error/loss per epoch to give a sense of convergence speed. Observe that SGP with SLOWMO substantially improves convergence, compared to SGP alone. We observe a similar phenomenon when comparing the training curves; see Appendix B.

Table 1: Comparisons to the original distributed optimization algorithms on various training tasks. The best training loss, validation accuracy (for image classification), and BLEU score (for machine translation) are reported. We fix slow learning rate $\alpha = 1$. We set the number of local steps $\tau = 12$ for CIFAR10. For ImageNet and WMT, we use $\tau = 48$ for SGP and OSGP and $\tau = 12$ for Local SGD. The slow momentum $\beta$ is tuned for each case. It typically ranges from $0.4$ to $0.8$.

| Datasets | Baseline | Training Loss | | Validation Acc./BLEU | |
|---|---|---|---|---|---|
| | | Original | w/ SLOWMO | Original | w/ SLOWMO |
| CIFAR-10 | Local SGD | 0.122 | **0.006** | 91.73% | **93.20**% |
| | OSGP | 0.011 | **0.001** | 93.17% | **93.74**% |
| | SGP | 0.002 | **0.001** | 93.90% | **94.32**% |
| | AR-SGD | 0.002 | - | 92.66% | - |
| ImageNet | Local SGD | 1.43 | **1.21** | 69.94% | **73.24**% |
| | OSGP | 1.03 | **0.97** | 74.96% | **75.54**% |
| | SGP | 1.07 | **1.00** | 75.15% | **75.73**% |
| | AR-SGD | 0.96 | - | 76.00% | - |
| WMT'16 En-De | Local Adam | 2.520 | **2.480** | 26.62 | **27.14** |
| | SGP | 2.500 | **2.447** | 26.92 | **27.84** |
| | AR-Adam | 2.468 | - | 27.17 | - |

**Communication Cost.** Table 2 shows the average training time per iteration on ImageNet and WMT'16. For SGP/OSGP, since the additional communication cost due to averaging in line 6 of Algorithm 1 is amortized over $\tau = 48$ iterations, SLOWMO maintains nearly the same speed as the corresponding base algorithm. For methods like Local SGD or Local Adam, which already compute an exact average every $\tau$ iterations, using SlowMo (*i.e.*, using $\beta > 0$) does not increase the amount of communication. In other words, using SLOWMO on top of the base algorithm improves training/validation accuracy at a negligible additional communication cost.

**Effects of $\tau$.** The most important hyper-parameter in SLOWMO is the number of base optimizer steps $\tau$ before each SLOWMO update, since it influences both the accuracy and the training time. Figure 3 presents the validation accuracy and average iteration time of SGP-SLOWMO for different values of $\tau$ on ImageNet and WMT'16. It can be observed that the validation performance does not monotonically increase or decrease with $\tau$. Instead, there is a best value. On both ImageNet and WMT'16, we find $\tau = 48$ to be a good tradeoff between speed and accuracy. Moreover, SLOWMO is pretty robust to the choice of $\tau$; even if $\tau = 96$ for ImageNet and $\tau = 192$ for WMT'16, SGP with SLOWMO achieves better validation accuracy/loss than SGP alone.

We further investigate the effect of other hyperparameters (the slow learning rate $\alpha$, slow momentum $\beta$) as well as the different strategies for handling base algorithm buffers in Appendix B.

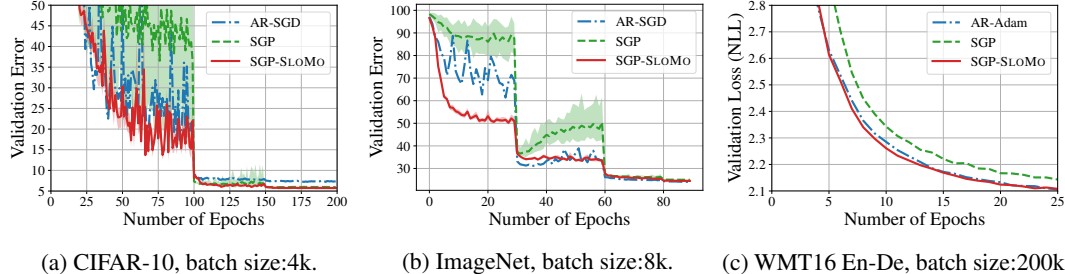

(a) CIFAR-10, batch size:4k.    (b) ImageNet, batch size:8k.    (c) WMT16 En-De, batch size:200k.

Figure 2: Validation curves for various tasks using SGP as the base algorithm. We fix $\alpha = 1, \tau = 12$ for these three plots. Shaded areas in (a) and (b) show the min-max values across all worker nodes. The corresponding training curves are presented in Appendix B.2.

Table 2: Average time per iteration with and without SLOWMO. Recall that $\tau = 48$ for the SGP and OSGP base optimizer and $\tau = 12$ for Local SGD/Local Adam. In some cases, with SLOWMO was faster than without; we hypothesize that this is due to statistical variations in timing and background network traffic.



(a) ImageNet, batch size:8k, 32 nodes.

| Baseline | Time/iterations (ms) | |
|---|---|---|
| | Original | w/ SLOWMO |
| Local SGD | 294 | 282 |
| OSGP | 271 | 271 |
| SGP | 304 | 302 |
| AR-SGD | 420 | - |

(b) WMT'16 En-De, batch size:200k, 8 nodes.

| Baseline | Time/iterations (ms) | |
|---|---|---|
| | Original | w/ SLOWMO |
| Local Adam | 503 | 505 |
| SGP | 1225 | 1279 |
| AR-Adam | 1648 | - |



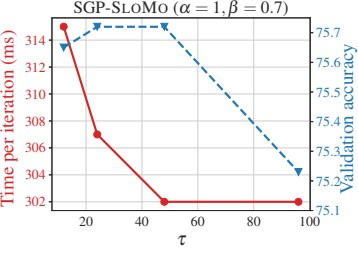

(a) Effect of $\tau$ on ImageNet.

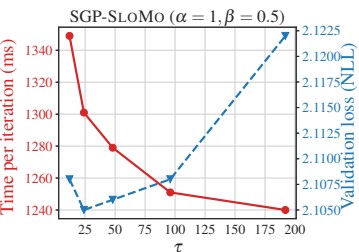

(b) Effect of $\tau$ on WMT'16.

Figure 3: The effects of $\tau$ in SLOWMO. We use SGP as the base algorithm. For ImageNet we plot validation accuracy (higher is better), and for WMT'16 En-De we plot validation NLL (lower is better). Increasing $\tau$ amortizes communication cost over more iterations, so the average time per iteration decreases. We hypothesize that moderate values of $\tau$ have a regularizing effect, improving loss and accuracy, and when $\tau$ is too large performance is degraded because workers' local models drift too far apart.

**Comparison with Double-Averaging Momentum.** As mentioned in Section 3, Yu et al. (2019a) propose an alternative momentum scheme, double-averaging, to improve the convergence of Local SGD and D-PSGD. We empirically compare it with SLOWMO in terms of the validation accuracy and average training time per iteration on ImageNet. When the base algorithm is SGP, double averaging achieves 75.54% validation accuracy and takes 402 ms per iteration on average, while SLOWMO-SGP ($\tau = 48$) reaches 75.73% validation accuracy while taking 302 ms per iteration on average. Similarly, when the baseline algorithm is Local SGD with $\tau = 12$, double-averaging reaches 72.04% and takes 405 ms per iteration, while SLOWMO reaches 73.24% and takes only 282 ms per iteration.

## 5 THEORETICAL RESULTS

This section provides a convergence guarantee for SLOWMO and shows that it can achieve a linear speedup in terms of number of workers. Let $f_i(\boldsymbol{x}) = \mathbb{E}_{\xi_i \sim D_i}[F_i(\boldsymbol{x}; \xi_i)]$ denote the expected objective function at worker $i$, and let $f(\boldsymbol{x}) = \frac{1}{m}\sum_{i=1}^{m} f_i(\boldsymbol{x})$. Our analysis is conducted for a constant learning rate $\gamma_t = \gamma$ under the following standard assumptions.

**Assumption 1** (*$L$-smooth*). *Each local objective function $f_i(\boldsymbol{x})$ is $L$-smooth, i.e., $\|\nabla f_i(\boldsymbol{x}) - \nabla f_i(\boldsymbol{y})\| \leq L \|\boldsymbol{x} - \boldsymbol{y}\|$, for all $\boldsymbol{x}, \boldsymbol{y} \in \mathbb{R}^d$ and $i \in \{1, 2, \ldots, m\}$.*

**Assumption 2** (*Bounded variance*). *There exists a finite positive constant $\sigma^2$ such that $\mathbb{E}_{\xi \sim D_i} \|\nabla F_i(\boldsymbol{x}; \xi) - \nabla f_i(\boldsymbol{x})\|^2 \leq \sigma^2$, for all $i \in \{1, 2, \ldots, m\}$.*

In order to generalize the analysis to various base algorithms, we define $\boldsymbol{d}_{t,k} = \frac{1}{m}\sum_{i=1}^{m} \boldsymbol{d}_{t,k}^{(i)}$ as the average descent direction across the $m$ workers and make the following assumption.

**Assumption 3.** *There exists a finite positive constant $V$ such that $\mathbb{E} \left\| \boldsymbol{d}_{t,k} - \mathbb{E}_{t,k}[\boldsymbol{d}_{t,k}] \right\|^2 \le V$, where $\mathbb{E}_{t,k}$ denotes expectation conditioned on all randomness from stochastic gradients up to the $k$-th step of $t$-th outer iteration.*

As mentioned in Section 2, the analytic form of $\boldsymbol{d}_{t,k}$ depends on the choice of base algorithm. Therefore, the value of $V$ also changes. For instance, when the base algorithm is Local-SGD, then $\boldsymbol{d}_{t,k} = \frac{1}{m} \sum_{i=1}^m \nabla F_i(\boldsymbol{x}_{t,k}^{(i)}; \xi_{t,k}^{(i)})$. It follows that

$$\mathbb{E} \left\| \boldsymbol{d}_{t,k} - \mathbb{E}_{t,k}[\boldsymbol{d}_{t,k}] \right\|^2 = \frac{1}{m^2} \sum_{i=1}^m \mathbb{E} \left\| \nabla F_i(\boldsymbol{x}_{t,k}^{(i)}; \xi_{t,k}^{(i)}) - \nabla f_i(\boldsymbol{x}_{t,k}^{(i)}) \right\|^2 \le \frac{\sigma^2}{m} = V. \tag{4}$$

The above value ($V = \sigma^2/m$) can also be applied to other base algorithms, such as D-PSGD, SGP, and OSGP. More details are provided in Appendix C.

Our main convergence result is stated next. Proofs of all results in this section appear in Appendix D.

**Theorem 1 (General Result).** *Suppose all worker nodes start from the same initial point $\boldsymbol{x}_{0,0}$, and the initial slow momentum is $\boldsymbol{u}_0 = \boldsymbol{0}$. If we set $\alpha$, $\beta$, $\gamma_t = \gamma$, $\tau$ and $T$ so that $\frac{\alpha\gamma}{1-\beta} = \sqrt{\frac{m}{\tau T}}$ and the total iterations $\tau T$ satisfies $\tau T \ge mL^2 \left( 1 + \sqrt{3} \max \left\{ \frac{3\tau(1-\beta-\alpha)}{\alpha}, \frac{4\tau\beta}{1-\beta}, 1 \right\} \right)$, then under Assumptions 1 to 3, we have that:*

$$\frac{1}{\tau T} \sum_{t=0}^{T-1} \sum_{k=0}^{\tau-1} \mathbb{E} \left\| \nabla f(\boldsymbol{x}_{t,k}) \right\|^2 \le \frac{2 \left( f(\boldsymbol{x}_{0,0}) - f_{inf} \right) + mVL}{\sqrt{m\tau T}} + \underbrace{\frac{1}{\tau T} \sum_{t=0}^{T-1} \sum_{k=0}^{\tau-1} \mathbb{E} \left\| \nabla f(\boldsymbol{x}_{t,k}) - \mathbb{E}_{t,k}[\boldsymbol{d}_{t,k}] \right\|^2}_{\textit{Effect of base optimizer}}$$

$$+ \underbrace{\frac{4mVL^2(\tau - 1)}{\tau T} \left( \frac{1-\beta}{\alpha} - 1 \right)^2 + \frac{8mVL^2\tau}{\tau T} \frac{\beta^2}{(1-\beta^2)}}_{\textit{Effect of slow momentum}} \tag{5}$$

*where $f_{inf} = \inf_{\boldsymbol{x}} f(\boldsymbol{x})$.*

**Consistent with AR-SGD.** Recall that AR-SGD is equivalent to taking $\tau = 1$, $\alpha = 1$, and $\beta = 0$ and using SGD with learning rate $\gamma$ as the base optimizer. In this case, all terms on the RHS but the first one vanish, $V = \sigma^2/m$, and (5) is identical to the well-known rate of $\mathcal{O}(1/\sqrt{mT\tau})$ for SGD.

**Effect of the base optimizer.** The second term in (5) only depends on the base optimizer. It measures the bias between the full batch gradient $\nabla f(\boldsymbol{x}_{t,k})$ and the expected update averaged across all workers $\mathbb{E}_{t,k}[\boldsymbol{d}_{t,k}]$. For the base optimizers considered in this paper, this term relates to the discrepancies among local models and can be easily found in previous distributed optimization literature. In particular, under the same assumptions as Theorem 1, one can show that this term vanishes in a rate of $1/(T\tau)$ for D-PSGD, SGP, OSGP and Local-SGD; see Appendix C.

As an example, we provide the convergence analysis for the extreme case of Local SGD, where there is no communication between nodes during each inner iteration. Intuitively, using other base algorithms should only make this term smaller since they involve more communication than Local SGD.

**Corollary 1 (Convergence of BMUF, *i.e.*, Local SGD with SLOWMO).** *Under the same conditions as Theorem 1, if the inner algorithm is Local SGD and there exists a positive finite constant $\zeta$ such that $\frac{1}{m} \sum_{i=1}^m \|\nabla f(\mathbf{x}) - \nabla f_i(\mathbf{x})\|^2 \le \zeta^2$, then*

$$\frac{1}{\tau T} \sum_{t=0}^{T-1} \sum_{k=0}^{\tau-1} \mathbb{E} \left\| \nabla f(\boldsymbol{x}_{t,k}) \right\|^2 = \mathcal{O} \left( \frac{1}{\sqrt{m\tau T}} \right) + \mathcal{O} \left( \frac{m\tau}{T} \right). \tag{6}$$

**Linear speedup.** Corollary 1 shows that when the total number of steps $T\tau$ is sufficiently large: $T \ge m^3\tau^3$, the convergence rate will be dominated by the first term $\mathcal{O}(1/\sqrt{mT\tau})$. That is, in order to achieve an $\epsilon$ error, the algorithm requires $m$ times less steps when using $m$ times more worker nodes. This also recovers the same rate as AR-SGD.

**Extension to single-node case.** As mentioned in Section 2, when there is only one node and the slow momentum factor is $\beta = 0$, the SLOWMO-SGD is the Lookahead optimizer. One can directly apply Theorem 1 to this special case and get the following corollary.

**Corollary 2 (Convergence of Lookahead).** *Under the same conditions as Theorem 1, if the inner optimizer is AR-SGD and $\beta = 0$, then one can obtain the following upper bound:*

$$\frac{1}{\tau T} \sum_{t=0}^{T-1} \sum_{k=0}^{\tau-1} \mathbb{E} \left\| \nabla f(\boldsymbol{x}_{t,k}) \right\|^2 \leq \frac{2 \left( f(\boldsymbol{x}_{0,0}) - f_{inf} \right) + \sigma^2 L}{\sqrt{\tau T}} + \frac{4 \sigma^2 L^2 (\tau - 1)}{\tau T} \left( \frac{1}{\alpha} - 1 \right)^2 \quad (7)$$

$$= \mathcal{O} \left( \frac{1}{\sqrt{\tau T}} \right) + \mathcal{O} \left( \frac{1}{T} \right). \quad (8)$$

## 6 FASTER SLOWMO: REMOVING THE PERIODIC ALLREDUCE

SLOWMO helps improve both the optimization and generalization of communication-efficient algorithms. When the base optimizer is SGP or OSGP, SLOWMO also comes at the expense of higher communication cost, since it requires performing an exact average every $\tau$ iterations. [1] Although the communication cost can be amortized, here we go one step further and propose a SGP-SLOWMO variant, named SGP-SLOWMO-noaverage, where we remove the exact average when we perform the SLOWMO update, i.e we skip line 6 in Algorithm 1. We empirically evaluate this variant on the ImageNet and WMT'16 datasets, using $\alpha = 1$, $\beta = 0.6$ and $\tau = 48$.

Surprisingly, we observe that SGP-SLOWMO-noaverage achieves similar performances on Imagenet (75.78%, compared to 75.73% for SGP-SLOWMO) and only slightly degrades the validation NLL on WMT'16 (2.11, compared to 2.10), while preserving the iteration time of the base algorithm (298 ms per iteration on ImageNet and 1227 ms per iteration on WMT'16) since this variant does not require additional communication. These results suggest that the slow momentum updates, and not the momentum buffer synchronization, contribute the most to the performance gain of SLOWMO. We leave further investigation of SLOWMO-SGP-noaverage for future work.

## 7 CONCLUDING REMARKS

In this paper, we propose a general momentum framework, SLOWMO, for communication-efficient distributed optimization algorithms. SLOWMO can be built on the top of SGD, as well as decentralized methods, such as SGP and (asynchronous) OSGP. On three different deep learning tasks, we empirically show that SLOWMO consistently improves the optimization and generalization performance of the corresponding baseline algorithm while maintaining a similar level of communication efficiency. Moreover, we establish a convergence guarantee for SLOWMO, showing that it converges to a stationary point of smooth and non-convex objective functions. Since BMUF (Chen & Huo, 2016) can be expressed through SLOWMO framework (by setting the base optimizer to be Local SGD or Local Adam), to the best of our knowledge, we provide the first convergence guarantee for BMUF in the literature.

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

# A  EXPERIMENT DETAILS

## A.1  IMPLEMENTATION DETAILS

All methods are implemented in PyTorch 1.0 (Paszke et al., 2017), and our experiments use CUDA 9.2, CUDNN 7.3, and NCCL 2.2.13. The ImageNet experiments build on the example from `https://github.com/pytorch/examples/imagenet`. The WMT'16 En-De experiments build on `https://github.com/pytorch/fairseq`. For SGP and OSGP we use the implementations available at `https://github.com/facebookresearch/stochastic_gradient_push`.

## A.2  CIFAR-10

For the CIFAR-10 experiments, we train a ResNet-18, the implementation of which is available at `https://github.com/kuangliu/pytorch-cifar/blob/master/models/resnet.py`. In all base algorithms, we use a Nesterov momentum parameter of 0.9 and set the weight decay factor as $10^{-4}$. For each base algorithm, we tune the (fast) learning rate from $\{0.01, 0.025, 0.05, 0.1, 0.15\}$ and linearly scale it with the number of workers (*i.e.*, 32). We found that, with a total batch size 4096, the best learning rate for AR-SGD is 0.01, for OSGP/SGP is 0.05, and for Local SGD is 0.025.

When applying SLOWMO to these base algorithms, we fix $\alpha = 1$ and $\tau = 12$ and tune the value of $\beta$ from $\{0.4, 0.5, 0.6, 0.7, 0.8\}$. It turns out that for OSGP, SGP, and Local SGD, the best values of $\beta$ are all equal to 0.7. More discussion on the effects of $\alpha$ and $\beta$ can be found in Appendix B.3.

## A.3  IMAGENET

For the ImageNet experiments, we use the same learning-rate, schedule, momentum, and weight decay as those suggested in Goyal et al. (2017) for SGD. In particular, we use ResNet50 (He et al. (2016)) and train it for 90 epochs with a reference learning-rate of 0.1 with respect to a 256 sample batch, and scale this linearly with the batch-size. We decay the learning-rate by a factor of 10 at epochs 30, 60, 80. We use a Nesterov momentum parameter of 0.9. We use weight decay $10^{-4}$.

When using SLOWMO, we set the slow learning rate to $\alpha = 1$ and explore different numbers of inner steps, $\tau \in \{12, 48\}$ and different slow momentum value $\beta \in \{0.5, 0.6, 0.7\}$ when the base optimizer is SGP/OSGP and $\beta = 0.7$ when the base optimizer is LocalSGD. We also explore a larger set of $\tau$ values $\tau \in \{12, 24, 48, 96\}$ in the ablation experiments.

## A.4  WMT16 EN-DE

For the WMT16 En-De experiments, we follow the experimental protocol described in Ott et al. (2018). All experiments are based on the big transformer model (Vaswani et al., 2017) with 6 blocks in the encoder and decoder networks. For these experiments, the base optimizer used is Adam (Kingma & Ba, 2015) with beta1 = 0.9, beta2 = 0.98, and $\epsilon = 10^{-8}$ and trained for 25 epochs. We use the same learning rate schedule as Ott et al. (2018), *i.e.*, the learning rate increases linearly for $4,000$ steps to $10^{-3}$, after which it is decayed proportionally to the inverse square root of the number of steps. We use label smoothing with weight 0.1 for the uniform prior distribution.

For SLOWMO, we explore $\{0.5, 1.0\}$ as the slow learning rate $\alpha$. We observe that $\alpha = 1$ gives better performance and therefore report results for $\alpha = 1$ unless stated otherwise. We explore different numbers of inner steps, $\tau \in \{12, 48\}$ and different slow momentum value $\beta \in \{0.1, 0.3, 0.5, 0.6, 0.7\}$. We also explore a larger set of $\tau$ values, i.e. $\tau \in \{12, 24, 48, 96, 192\}$, in the ablation experiments.

Table B.1: Validation NLL (lower is better) with and without SLOWMO on WMT'16 En-De. We observe that SLOWMO improves the validation NLL of SGP and Local Adam.

| Baseline | Validation NLL | |
| --- | --- | --- |
| | Original | w/ SLOWMO |
| Local Adam | 2.179 | **2.137** |
| SGP | 2.137 | **2.106** |
| AR-Adam | 2.108 | - |

## B ADDITIONAL EMPIRICAL RESULTS

### B.1 VALIDATION NLL ON WMT'16 EN-DE

We show the Validation NLL on WMT'16 En-De in Table B.1, corresponding to the experiments in Table 1. We observe that SLOWMO improves the validation NLL (along with BLEU score) of SGP and Local Adam.

### B.2 ADDITIONAL TRAINING CURVES

We present the training loss-versus-epochs curves in Figure B.1, corresponding to the validation curves in Figure 2. It can be observed that SLOWMO substantially improves the convergence speed of SGP.

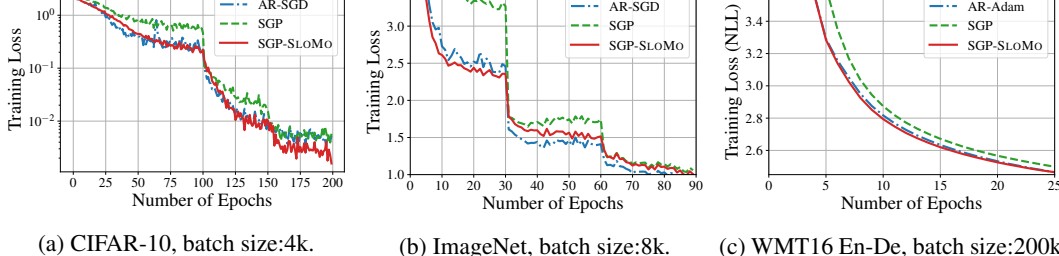

(a) CIFAR-10, batch size:4k.  (b) ImageNet, batch size:8k.  (c) WMT16 En-De, batch size:200k.

Figure B.1: Training curves for various tasks using SGP as the base algorithm. We fix $\alpha = 1, \tau = 12$ for these three plots. Shaded areas in (a) and (b) show the min-max values across all worker nodes. The corresponding validation curves are presented in Figure 2. Note that the training losses in these three figures are evaluated right after the SLOWMO update (*i.e.*, Eq. (3)).

### B.3 EFFECT OF SLOW LEARNING RATE $\alpha$ AND SLOW MOMENTUM FACTOR $\beta$

In this section we evaluate the impact of slow learning rate $\alpha$ and slow momentum $\beta$ hyperameters.

In Figure B.2a, we perform a parameter sweep over $\alpha$ and $\beta$ on CIFAR-10 dataset, using OSGP as the base algorithm of SLOWMO. One can observe that when the value of $\beta$ is fixed, $\alpha = 1$ always gives the highest validation accuracy; when the value of $\alpha$ is fixed, there is a best value of $\beta$ ranging from $0.4$ to $0.8$.

We further validate this claim on the WMT'16-En-De dataset. Figure B.2b shows that $\alpha = 1$ gives lower validation loss than $\alpha = 0.5$ for fixed $\beta$ when using SGP or Local Adam as the base algorithms. When running SLOWMO-Adam with $\beta > 0.5$ and $\alpha = 1.0$, or with $\beta > 0.7$ and $\alpha = 0.5$, the validation loss was substantially worse and so is not plotted here. Motivated by the above observations, we stick to fix $\alpha = 1$ and fine-tune $\beta$ for SLOWMO methods on all training tasks.

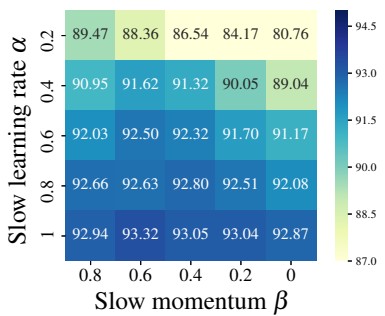

(a) Parameter sweep on CIFAR-10 training task using OSGP as the base algorithm.

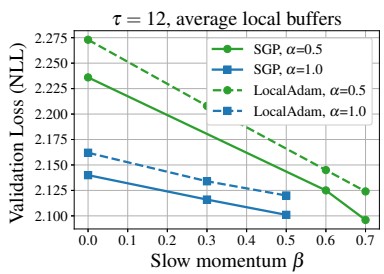

(b) Effect of $\alpha$ and $\beta$ on WMT'16 on SLOWMO-Adam.

Figure B.2: Impact on slow learning rate $\alpha$ and slow momentum $\beta$ on SLOWMO.

## B.4 BASE OPTIMIZER MOMENTUM BUFFER STRATEGIES

As described in Section 2, the base optimizer may have some associated buffers. SGD with momentum uses a momentum buffer, and Adam tracks estimates of the first and second moments of the gradient. The slow momentum buffer is updated every $\tau$ steps according to Eq.(2). There are several strategies that can be used to update the base optimizer buffers at the outer iteration level (line 2 in Algorithm 1). Here, we explore three strategies: 1) *reset* the base optimizer buffers to zero ; 2) *maintain* the base optimizer buffers to their current values; 3) *average* the base optimizer buffers across workers, which requires additional communications. We evaluate the impact of these strategies on ImageNet and WMT'16 in Table B.2 and Table B.3.

On ImageNet, we observe that the different buffer strategies achieve similar training and validation performance. However, the averaging strategy comes at the cost of higher communication overhead (an additional call to ALLREDUCE for each buffer averaged). Based on these results, we choose the reset strategy as the default in our experiments.

On WMT'16, we find that the reset buffer strategy underperforms both the maintain and average approaches. When using Adam as base optimizer, resetting the second moment to zeros hurts the optimization performance. This is not surprising since it is recognized that warming up the Adam buffer is important. Averaging buffers achieves the best results but comes at a significantly higher communication cost. We therefore select the maintain strategy as the default one when using Adam.

Table B.2: Effect of different buffer strategies: ImageNet, batch size:8k, 32 nodes.

| Buffer Strategy | Training Loss | Validation Accuracy |
|---|---|---|
| Avg parameters + avg buffers | 1.06 | 75.66% |
| Avg parameters + reset buffers | 1.00 | 75.73% |
| Avg parameters + maintain buffers | 0.98 | 75.78% |

Table B.3: Effect of different buffer strategies: WMT'16 En-De, batch size:200k, 8 nodes.

| Buffer Strategy | Training Loss | Validation Loss |
|---|---|---|
| Avg parameters + avg buffers | 2.438 | 2.101 |
| Avg parameters + reset buffers | 5.093 | 4.732 |
| Avg parameters + maintain buffers | 2.447 | 2.106 |

## B.5 STANDARD DEVIATIONS ON CIFAR-10

Since experiments for CIFAR-10 were ran for 5 times with different random seeds, here we report the standard deviations on the validation accuracy in Table B.4, as a complementary to Table 1.

Table B.4: Validation Accuracy with and without SLOWMO on CIFAR-10. Using SLOWMO consistently improves the performance of the base algorithms.

| Baseline | Validation Accuracy | |
| --- | --- | --- |
| | Original | w/ SLOWMO |
| Local SGD | $91.73 \pm .14\%$ | $\mathbf{93.20 \pm .23}\%$ |
| OSGP | $93.17 \pm .11\%$ | $\mathbf{93.74 \pm .17}\%$ |
| SGP | $93.90 \pm .13\%$ | $\mathbf{94.32 \pm .21}\%$ |
| AR-SGD | $92.66 \pm .16\%$ | - |

## C  BASELINE ALGORITHMS

In this section, we give a detailed description of each baseline algorithms used throughout the paper, provide theoretical justification on how to incorporate the update rules of D-PSGD, SGP and OSGP into the analysis of SLOWMO, and also derive the analytic form of $V$ used in Assumption 3 for each method. A summary of the base optimizer update directions is given in Table C.1

Table C.1: Examples of update directions used by the base optimizer in SLOWMO, where $\boldsymbol{h}^{(i)}$ and $\boldsymbol{v}^{(i)}$ denote the first-order and second-order momentum buffers respectively and $\beta_{\text{local}}, \beta_1, \beta_2$ are the corresponding local momentum factors. When the local momentum buffers are reset at the beginning of each inner loop, then $\boldsymbol{h}_{t,0}^{(i)} = 0, \boldsymbol{v}_{t,0}^{(i)} = 0$ and $l = k$; when the local momentum buffers are maintained, then $\boldsymbol{h}_{t,0}^{(i)} = \boldsymbol{h}_{t-1,\tau}^{(i)}, \boldsymbol{v}_{t,0}^{(i)} = \boldsymbol{v}_{t-1,\tau}^{(i)}$ and $l = t\tau + k$.

| Base Optimizer | Update directions $\boldsymbol{d}_{t,k}^{(i)}$(and possible additional buffers $\boldsymbol{h}^{(i)}, \boldsymbol{v}^{(i)}$) |
| --- | --- |
| SGD | $\boldsymbol{h}_{t,k+1}^{(i)} = \beta_{\text{local}}\boldsymbol{h}_{t,k}^{(i)} + \nabla F_i(\boldsymbol{x}_{t,k}^{(i)}; \xi_{t,k}^{(i)})$ 
 $\boldsymbol{d}_{t,k}^{(i)} = \beta_{\text{local}}\boldsymbol{h}_{t,k+1}^{(i)} + \nabla F_i(\boldsymbol{x}_{t,k}^{(i)}; \xi_{t,k}^{(i)})$ |
| SGP | $\boldsymbol{h}_{t,k+1}^{(i)} = \beta_{\text{local}}\boldsymbol{h}_{t,k}^{(i)} + \nabla F_i(\boldsymbol{z}_{t,k}^{(i)}; \xi_{t,k}^{(i)})$ 
 $\boldsymbol{d}_{t,k}^{(i)} = \frac{1}{\gamma_t}\boldsymbol{x}_{t,k}^{(i)} - \frac{1}{\gamma_t}\sum_{j\in\mathcal{N}_k^{in(i)}} p^{(i,j)}[\boldsymbol{x}_{t,k}^{(i)} - \gamma_t(\beta_{\text{local}}\boldsymbol{h}_{t,k+1}^{(i)} + \nabla F_i(\boldsymbol{z}_{t,k}^{(i)}; \xi_{t,k}^{(i)}))]$ |
| Adam | $\boldsymbol{h}_{t,k+1}^{(i)} = \beta_1\boldsymbol{h}_{t,k}^{(i)} + (1-\beta_1)\nabla F_i(\boldsymbol{x}_{t,k}^{(i)}; \xi_{t,k}^{(i)})$ 
 $\boldsymbol{v}_{t,k+1}^{(i)} = \beta_2\boldsymbol{v}_{t,k}^{(i)} + (1-\beta_2)\nabla F_i^2(\boldsymbol{x}_{t,k}^{(i)}; \xi_{t,k}^{(i)})$ 
 $\hat{\boldsymbol{h}}_{t,k+1}^{(i)} = \boldsymbol{h}_{t,k+1}^{(i)}/(1-\beta_1^l), \hat{\boldsymbol{v}}_{t,k+1}^{(i)} = \boldsymbol{v}_{t,k+1}^{(i)}/(1-\beta_2^l)$ 
 $\boldsymbol{d}_{t,k}^{(i)} = \hat{\boldsymbol{h}}_{t,k+1}^{(i)}/(\sqrt{\hat{\boldsymbol{v}}_{t,k+1}^{(i)}} + \epsilon)$ |
| SGP (Adam) | $\boldsymbol{h}_{t,k+1}^{(i)} = \beta_1\boldsymbol{h}_{t,k}^{(i)} + (1-\beta_1)\nabla F_i(\boldsymbol{z}_{t,k}^{(i)}; \xi_{t,k}^{(i)})$ 
 $\boldsymbol{v}_{t,k+1}^{(i)} = \beta_2\boldsymbol{v}_{t,k}^{(i)} + (1-\beta_2)\nabla F_i^2(\boldsymbol{z}_{t,k}^{(i)}; \xi_{t,k}^{(i)})$ 
 $\hat{\boldsymbol{h}}_{t,k+1}^{(i)} = \boldsymbol{h}_{t,k+1}^{(i)}/(1-\beta_1^l), \hat{\boldsymbol{v}}_{t,k+1}^{(i)} = \boldsymbol{v}_{t,k+1}^{(i)}/(1-\beta_2^l)$ 
 $\boldsymbol{d}_{t,k}^{(i)} = \frac{1}{\gamma_t}\boldsymbol{x}_{t,k}^{(i)} - \frac{1}{\gamma_t}\sum_{j\in\mathcal{N}_k^{in(i)}} p^{(i,j)}[\boldsymbol{x}_{t,k}^{(i)} - \gamma_t\hat{\boldsymbol{h}}_{t,k+1}^{(i)}/(\sqrt{\hat{\boldsymbol{v}}_{t,k+1}^{(i)}} + \epsilon)]$ |

### C.1  SGP AND OSGP

Algorithms 2 and 3 present the pseudo-code of SGP and OSGP (Assran et al., 2019). To be consistent with the experimental results of the original paper, we also use local Nesterov momentum for each worker node. The communication topology among worker nodes is a time-varying directed exponential graph Assran et al. (2019). That is, if all nodes are ordered sequentially, then, according to their rank $(0, 1, \ldots, m-1)$, each node periodically communicates with peers that are $2^0, 2^1, \ldots, 2^{\lfloor\log_2(m-1)\rfloor}$ hops away. We let each node only send and receive a single message (*i.e.*, communicate with 1 peer) at each iteration.

---

**Algorithm 2:** Stochastic Gradient Push with Nesterov Momentum (SGP)

---

**Input:** learning rate $\gamma$; momentum factor $\beta_0$; Number of worker nodes $m$. Initial point $\boldsymbol{x}_0^{(i)} = \boldsymbol{z}_0^{(i)}$, $\boldsymbol{h}_0^{(i)} = 0$ and $w_0^{(i)} = 1$ for all nodes $i \in \{1, 2, \ldots, m\}$.

1 **for** $k \in \{0, 1, \ldots, K-1\}$ **at worker** $i$ **in parallel do**

2     Compute mini-batch gradients: $\nabla F_i(\boldsymbol{z}_k^{(i)}; \xi_k^{(i)})$

3     Update local momentum: $\boldsymbol{h}_{k+1}^{(i)} = \beta_0 \boldsymbol{h}_k^{(i)} + \nabla F_i(\boldsymbol{z}_k^{(i)}; \xi_k^{(i)})$

4     $\boldsymbol{x}_{k+\frac{1}{2}}^{(i)} = \boldsymbol{x}_k^{(i)} - \gamma[\beta_0 \boldsymbol{h}_{k+1}^{(i)} + \nabla F_i(\boldsymbol{z}_k^{(i)}; \xi_k^{(i)})]$

5     Send $(p_k^{(j,i)} \boldsymbol{x}_{k+\frac{1}{2}}^{(i)}, p_k^{(j,i)} w_k^{(i)})$ to out-neighbors

6     Receive $(p_k^{(i,j)} \boldsymbol{x}_{k+\frac{1}{2}}^{(j)}, p_k^{(i,j)} w_k^{(j)})$ from in-neighbors

7     Update model parameters: $\boldsymbol{x}_{k+1}^{(i)} = \sum_{j \in \mathcal{N}_k^{in(i)}} p_k^{(i,j)} \boldsymbol{x}_{k+\frac{1}{2}}^{(j)}$

8     Update de-biased factors: $w_{k+1}^{(i)} = \sum_{j \in \mathcal{N}_k^{in(i)}} p_k^{(i,j)} w_{k+\frac{1}{2}}^{(j)}$

9     Update de-biased model parameters: $\boldsymbol{z}_{k+1}^{(i)} = \boldsymbol{x}_{k+1}^{(i)} / w_{k+1}^{(i)}$

10 **end**

---

**Algorithm 3:** Overlap Stochastic Gradient Push with Nesterov Momentum (OSGP)

---

**Input:** learning rate $\gamma$; momentum factor $\beta_0$; Number of worker nodes $m$. Initial point $\boldsymbol{x}_0^{(i)} = \boldsymbol{z}_0^{(i)}$, $\boldsymbol{h}_0^{(i)} = 0$ and $w_0^{(i)} = 1$ for all nodes $i \in \{1, 2, \ldots, m\}$; count_since_last = 0.

1 **for** $k \in \{0, 1, \ldots, K-1\}$ **at worker** $i$ **in parallel do**

2     Compute mini-batch gradients: $\nabla F_i(\boldsymbol{z}_k^{(i)}; \xi_k^{(i)})$

3     Update local momentum: $\boldsymbol{h}_{k+1}^{(i)} = \beta_0 \boldsymbol{h}_k^{(i)} + \nabla F_i(\boldsymbol{z}_k^{(i)}; \xi_k^{(i)})$

4     $\boldsymbol{x}_{k+\frac{1}{2}}^{(i)} = \boldsymbol{x}_k^{(i)} - \gamma[\beta_0 \boldsymbol{h}_{k+1}^{(i)} + \nabla F_i(\boldsymbol{z}_k^{(i)}; \xi_k^{(i)})]$

5     Non-blocking send $(p_k^{(j,i)} \boldsymbol{x}_{k+\frac{1}{2}}^{(i)}, p_k^{(j,i)} w_k^{(i)})$ to out-neighbors

6     $\boldsymbol{x}_{k+1}^{(i)} = p^{(i,i)} \boldsymbol{x}_k^{(i)}$

7     $w_{k+1}^{(i)} = p^{(i,i)} w_k^{(i)}$

8     **if** *count_since_last = s* **then**

9        Block until $(p_k^{(i,j)} \boldsymbol{x}_{k+\frac{1}{2}}^{(j)}, p_k^{(i,j)} w_k^{(j)})$ is received from in-neighbors

10        count_since_last = 0

11     **else**

12        count_since_last = count_since_last + 1

13     **end**

14     **if** *Receive buffer non-empty* **then**

15        **for** $(p_{k'}^{(j,i)} \boldsymbol{x}_{k'+\frac{1}{2}}^{(i)}, p_{k'}^{(j,i)} w_{k'}^{(i)})$ *in the receive buffer* **do**

16           $\boldsymbol{x}_{k+1}^{(i)} = \boldsymbol{x}_{k+1}^{(i)} + p_{k'}^{(i,j)} \boldsymbol{x}_{k'+\frac{1}{2}}^{(j)}$

17           $w_{k+1}^{(i)} = w_{k+1}^{(i)} + p_{k'}^{(i,j)} w_{k'+\frac{1}{2}}^{(j)}$

18        **end**

19     **end**

20     Update de-biased model parameters: $\boldsymbol{z}_{k+1}^{(i)} = \boldsymbol{x}_{k+1}^{(i)} / w_{k+1}^{(i)}$

21 **end**

---

Note that although the implementation of SGP is with Nesterov momentum, the theoretical analysis in Assran et al. (2019) only considers the vanilla case where there is no momentum. Accordingly, the update rule can be written in a matrix form as

$$\boldsymbol{X}_{k+1} = (\boldsymbol{X}_k - \gamma \nabla \boldsymbol{F}(\boldsymbol{Z}_k; \boldsymbol{\xi}_k)) \, \boldsymbol{P}_k^\top \tag{9}$$

where $\boldsymbol{X}_k = [\boldsymbol{x}_k^{(1)}, \ldots, \boldsymbol{x}_k^{(m)}] \in \mathbb{R}^{d \times m}$ stacks all model parameters at different nodes and $\boldsymbol{Z}_k = [\boldsymbol{z}_k^{(1)}, \ldots, \boldsymbol{z}_k^{(m)}] \in \mathbb{R}^{d \times m}$ denotes the de-biased parameters. Similarly, we define the stochastic gradient matrix as $\nabla \boldsymbol{F}(\boldsymbol{Z}_k) = [\nabla F_1(\boldsymbol{z}_k^{(1)}; \xi_k^{(i)}), \ldots, \nabla F_m(\boldsymbol{z}_k^{(i)}; \xi_k^{(i)})] \in \mathbb{R}^{d \times m}$. Moreover, $\boldsymbol{P}_k \in \mathbb{R}^{m \times m}$ is defined as the mixing matrix which conforms to the underlying communication topology. If node $j$ is one of the in-neighbors of node $i$, then $p^{(i,j)} > 0$ otherwise $p^{(i,j)} = 0$. In particular, matrix $\boldsymbol{P}_k$ is column-stochastic.

If we multiply a vector $\boldsymbol{1}/m$ on both sides of the update rule (9), we have

$$\boldsymbol{x}_{k+1} = \boldsymbol{x}_k - \frac{\gamma}{m} \sum_{i=1}^{m} \nabla F_i(\boldsymbol{z}_k^{(i)}; \xi_k^{(i)}) \tag{10}$$

where $\boldsymbol{x}_k = \boldsymbol{X}_k \boldsymbol{1}/m$ denotes the average model across all worker nodes. Recall that in SLOWMO, we rewrite the updates of the base algorithm at the $k$th steps of the $t$th outer iteration as

$$\boldsymbol{x}_{t,k+1} = \boldsymbol{x}_{t,k} - \gamma \boldsymbol{d}_{t,k}. \tag{11}$$

So comparing (10) and (11), we can conclude that $\boldsymbol{d}_{t,k} = \frac{1}{m} \sum_{i=1}^{m} \nabla F_i(\boldsymbol{z}_{t,k}^{(i)}; \xi_{t,k}^{(i)})$. As a consequence, we have $\mathbb{E}_{t,k}[\boldsymbol{d}_{t,k}] = \frac{1}{m} \sum_{i=1}^{m} \nabla f_i(\boldsymbol{z}_{t,k}^{(i)})$. Since mini-batch gradients are independent, it follows that

$$\mathbb{E} \|\boldsymbol{d}_{t,k} - \mathbb{E}_{t,k}[\boldsymbol{d}_{t,k}]\|^2 = \mathbb{E} \left\| \frac{1}{m} \sum_{i=1}^{m} \left[ \nabla F_i(\boldsymbol{z}_{t,k}^{(i)}; \xi_{t,k}^{(i)}) - \nabla f_i(\boldsymbol{z}_{t,k}^{(i)}) \right] \right\|^2 \tag{12}$$

$$= \frac{1}{m^2} \sum_{i=1}^{m} \mathbb{E} \left\| \nabla F_i(\boldsymbol{x}_{t,k}^{(i)}; \xi_{t,k}^{(i)}) - \nabla f_i(\boldsymbol{x}_{t,k}^{(i)}) \right\|^2 \tag{13}$$

$$\leq \frac{\sigma^2}{m} = V. \tag{14}$$

Similarly, for OSGP, one can repeat the above procedure again. But the definition of $\boldsymbol{X}_k, \boldsymbol{Z}_k$ and $\nabla \boldsymbol{F}(\boldsymbol{Z}_k)$ will change, in order to account for the delayed messages. In this case, we still have the update rule Eq. (10). But $\boldsymbol{x}_k$ is no longer the averaged model across all nodes. It also involves delayed model parameters. We refer the interested reader to Assran et al. (2019) for futher details.

## C.2 D-PSGD

In the case of *decentralized parallel SGD* (D-PSGD), proposed in Lian et al. (2017), the update rule is quite similar to SGP. However, the communication topology among worker nodes is an undirected graph. Hence, the mixing matrix $\boldsymbol{P}_k$ is doubly-stochastic. Each node will exchange the model parameters with its neighbors. The update rule can be written as

$$\boldsymbol{X}_{k+1} = (\boldsymbol{X}_k - \gamma \nabla \boldsymbol{F}(\boldsymbol{X}_k; \boldsymbol{\xi}_k)) \boldsymbol{P}_k^\top. \tag{15}$$

Again, we have that $\boldsymbol{X}_k = [\boldsymbol{x}_k^{(1)}, \ldots, \boldsymbol{x}_k^{(m)}] \in \mathbb{R}^{d \times m}$ stacks all model parameters at different nodes and $\nabla \boldsymbol{F}(\boldsymbol{X}_k) = [\nabla F_1(\boldsymbol{x}_k^{(1)}; \xi_k^{(i)}), \ldots, \nabla F_m(\boldsymbol{x}_k^{(i)}; \xi_k^{(i)})] \in \mathbb{R}^{d \times m}$ denotes the stochastic gradient matrix. By multiplying a vector $\boldsymbol{1}/m$ on both sides of (15), we have

$$\boldsymbol{x}_{k+1} = \boldsymbol{x}_k - \frac{\gamma}{m} \sum_{i=1}^{m} \nabla F_i(\boldsymbol{x}_k^{(i)}; \xi_k^{(i)}) \tag{16}$$

$$= \boldsymbol{x}_k - \gamma \boldsymbol{d}_k. \tag{17}$$

As a result, using the same technique as (12)-(14), we have $V = \sigma^2/m$ for D-PSGD.

## C.3 LOCAL SGD

We further present the pseudo-code of Local SGD in Algorithm 4, and the pseudo-code of double-averaging momentum scheme in Algorithm 5.

---

**Algorithm 4:** Local SGD with Nesterov Momentum

---

**Input:** learning rate $\gamma$; momentum factor $\beta_0$; Communication period $\tau$; Number of worker nodes $m$. Initial point $\boldsymbol{x}_0^{(i)}$ and $\boldsymbol{h}_0^{(i)} = 0$ for all nodes $i \in \{1, 2, \ldots, m\}$.

1 **for** $k \in \{0, 1, \ldots, K-1\}$ **at worker** $i$ **in parallel do**
2 $\quad$ Compute mini-batch gradients: $\nabla F_i(\boldsymbol{x}_k^{(i)}; \xi_k^{(i)})$
3 $\quad$ Update local momentum: $\boldsymbol{h}_{k+1}^{(i)} = \beta_0 \boldsymbol{h}_k^{(i)} + \nabla F_i(\boldsymbol{x}_k^{(i)}; \xi_k^{(i)})$
4 $\quad$ $\boldsymbol{x}_{k+\frac{1}{2}}^{(i)} = \boldsymbol{x}_k^{(i)} - \gamma[\beta_0 \boldsymbol{h}_{k+1}^{(i)} + \nabla F_i(\boldsymbol{z}_k^{(i)}; \xi_k^{(i)})]$
5 $\quad$ **if** $k \bmod \tau = 0$ **then**
6 $\quad\quad$ ALLREDUCE model parameters: $\boldsymbol{x}_{k+1}^{(i)} = \frac{1}{m} \sum_{j=1}^{m} \boldsymbol{x}_{k+\frac{1}{2}}^{(j)}$
7 $\quad$ **else**
8 $\quad\quad$ $\boldsymbol{x}_{k+1}^{(i)} = \boldsymbol{x}_{k+\frac{1}{2}}^{(i)}$
9 $\quad$ **end**
10 **end**

---

**Algorithm 5:** Local SGD with Double-Averaging Nesterov Momentum (Yu et al., 2019a)

---

**Input:** learning rate $\gamma$; momentum factor $\beta_0$; Communication period $\tau$; Number of worker nodes $m$. Initial point $\boldsymbol{x}_0^{(i)}$ and $\boldsymbol{h}_0^{(i)} = 0$ for all nodes $i \in \{1, 2, \ldots, m\}$.

1 **for** $k \in \{0, 1, \ldots, K-1\}$ **at worker** $i$ **in parallel do**
2 $\quad$ Compute mini-batch gradients: $\nabla F_i(\boldsymbol{x}_k^{(i)}; \xi_k^{(i)})$
3 $\quad$ Update local momentum: $\boldsymbol{h}_{k+\frac{1}{2}}^{(i)} = \beta_0 \boldsymbol{h}_k^{(i)} + \nabla F_i(\boldsymbol{x}_k^{(i)}; \xi_k^{(i)})$
4 $\quad$ $\boldsymbol{x}_{k+\frac{1}{2}}^{(i)} = \boldsymbol{x}_k^{(i)} - \gamma[\beta_0 \boldsymbol{h}_{k+\frac{1}{2}}^{(i)} + \nabla F_i(\boldsymbol{z}_k^{(i)}; \xi_k^{(i)})]$
5 $\quad$ **if** $k \bmod \tau = 0$ **then**
6 $\quad\quad$ ALLREDUCE model parameters: $\boldsymbol{x}_{k+1}^{(i)} = \frac{1}{m} \sum_{j=1}^{m} \boldsymbol{x}_{k+\frac{1}{2}}^{(j)}$
7 $\quad\quad$ ALLREDUCE momentum buffers: $\boldsymbol{h}_{k+1}^{(i)} = \frac{1}{m} \sum_{j=1}^{m} \boldsymbol{h}_{k+\frac{1}{2}}^{(j)}$
8 $\quad$ **else**
9 $\quad\quad$ $\boldsymbol{x}_{k+1}^{(i)} = \boldsymbol{x}_{k+\frac{1}{2}}^{(i)}$
10 $\quad\quad$ $\boldsymbol{h}_{k+1}^{(i)} = \boldsymbol{h}_{k+\frac{1}{2}}^{(i)}$
11 $\quad$ **end**
12 **end**

---

## D PROOFS

### D.1 EQUIVALENT UPDATES

To begin, recall that $\boldsymbol{d}_{t,k} = \frac{1}{m} \sum_{i=1}^{m} \boldsymbol{d}_{t,k}^{(i)}$, and that the local updates are

$$\boldsymbol{x}_{t,k+1}^{(i)} = \boldsymbol{x}_{t,k}^{(i)} - \gamma \boldsymbol{d}_{t,k}^{(i)} \tag{18}$$

for $k = 0, \ldots, \tau - 1$, followed by an averaging step to obtain $\boldsymbol{x}_{t,\tau} = \frac{1}{m} \sum_{i=1}^{m} \boldsymbol{x}_{t,\tau}^{(i)}$. Therefore, we can write the update rule of the base optimizer as

$$\boldsymbol{x}_{t,\tau} = \boldsymbol{x}_{t,0} - \gamma \sum_{k=0}^{\tau-1} \boldsymbol{d}_{t,k}. \tag{19}$$

Combining this with (2) and (3), we have

$$\boldsymbol{x}_{t+1,0} = \boldsymbol{x}_{t,0} - \alpha\gamma \sum_{k=0}^{\tau-1} \boldsymbol{d}_{t,k} - \alpha\gamma\beta\boldsymbol{u}_t \tag{20}$$

$$= \boldsymbol{x}_{t,0} - \alpha\gamma \sum_{k=0}^{\tau-1} \boldsymbol{d}_{t,k} + \beta \left( \boldsymbol{x}_{t,0} - \boldsymbol{x}_{t-1,0} \right) \tag{21}$$

Let $\boldsymbol{y}_{t,0} = \boldsymbol{x}_{t,0} + \frac{\beta}{1-\beta}(\boldsymbol{x}_{t,0} - \boldsymbol{x}_{t-1,0}), \forall t$. Then by rearranging terms we get

$$\boldsymbol{y}_{t+1,0} = \boldsymbol{y}_{t,0} - \frac{\alpha\gamma}{1-\beta} \sum_{k=0}^{\tau-1} \boldsymbol{d}_{t,k}. \tag{22}$$

Now, let us further extend the auxiliary sequence to all values of $k \neq 0$ as follows:

$$\boldsymbol{y}_{t,k+1} = \boldsymbol{y}_{t,k} - \frac{\alpha\gamma}{1-\beta} \boldsymbol{d}_{t,k}. \tag{23}$$

It is easy to show that $\boldsymbol{y}_{t,\tau} = \boldsymbol{y}_{t+1,0}$. In the sequel, we will analyze the convergence of sequence $\{\boldsymbol{y}_{t,k}\}$ instead of $\{\boldsymbol{x}_{t,k}\}$.

## D.2 PRELIMINARIES

In the table below, we list all notations used in this paper.

Table D.1: List of notations.

| | |
|---|---|
| Global learning rate | $\alpha$ |
| Global momentum factor | $\beta$ |
| learning rate | $\gamma$ |
| Outer iteration length | $\tau$ |
| Total number of outer iterations | $T$ |
| Total number of steps | $K$ |
| Liptschiz constant | $L$ |
| Number of worker nodes | $m$ |

Throughout the theoretical analysis, we will repeatedly use the following facts:

- **Fact 1**: $\langle \mathbf{a}, \mathbf{b} \rangle = \frac{1}{2} \|\mathbf{a}\|^2 + \frac{1}{2} \|\mathbf{b}\|^2 - \frac{1}{2} \|\mathbf{a} - \mathbf{b}\|^2$;
- **Fact 2**: According to Young's Inequality, for any $a > 0$, we have

$$\pm \langle \mathbf{a}, \mathbf{b} \rangle \leq \frac{1}{2a} \|\mathbf{a}\|^2 + \frac{a}{2} \|\mathbf{b}\|^2. \tag{24}$$

- **Fact 3**: $\|\mathbf{a} + \mathbf{b}\|^2 \leq 2 \|\mathbf{a}\|^2 + 2 \|\mathbf{b}\|^2$;
- **Fact 4**: Suppose $\{a_i\}_{i=1}^k$ is a set of non-negative scalars and $s = \sum_{i=1}^k a_i$. Then according to Jensen's Inequality, we have

$$\left\| \sum_{i=1}^n a_i \mathbf{b}_i \right\|^2 = s^2 \cdot \left\| \sum_{i=1}^k \frac{a_i}{s} \mathbf{b}_i \right\|^2 \leq s^2 \cdot \sum_{i=1}^k \frac{a_i}{s} \|\mathbf{b}_i\|^2 = s \cdot \sum_{i=1}^k a_i \|\mathbf{b}_i\|^2. \tag{25}$$

## D.3 GENERAL TREATMENT

Since each local objective $f_i$ is $L$-smooth, the function $f = \frac{1}{m} \sum_{i=1}^m f_i$ is also $L$-smooth. It follows that

$$\mathbb{E}_{t,k}[f(\boldsymbol{y}_{t,k+1})] - f(\boldsymbol{y}_{t,k}) \leq - \frac{\alpha\gamma}{1-\beta} \langle \nabla f(\boldsymbol{y}_{t,k}), \mathbb{E}_{t,k}[\boldsymbol{d}_{t,k}] \rangle + \frac{\alpha^2\gamma^2 L}{2(1-\beta)^2} \mathbb{E}_{t,k} \left[ \|\boldsymbol{d}_{t,k}\|^2 \right] \tag{26}$$

where $\mathbb{E}_{t,k}$ denotes a conditional expectation over the randomness in the $(t,k)$-th iteration, conditioned on all past random variables. For the first term on the right hand side:

$$
\begin{aligned}
-\langle \nabla f(\boldsymbol{y}_{t,k}),\, \mathbb{E}_{t,k}[\boldsymbol{d}_{t,k}]\rangle = & -\langle \nabla f(\boldsymbol{y}_{t,k}) - \nabla f(\boldsymbol{x}_{t,k}),\, \mathbb{E}_{t,k}[\boldsymbol{d}_{t,k}]\rangle - \langle \nabla f(\boldsymbol{x}_{t,k}),\, \mathbb{E}_{t,k}[\boldsymbol{d}_{t,k}]\rangle \\
\leq & \frac{1}{2a}\left\|\nabla f(\boldsymbol{y}_{t,k}) - \nabla f(\boldsymbol{x}_{t,k})\right\|^2 + \frac{a}{2}\left\|\mathbb{E}_{t,k}[\boldsymbol{d}_{t,k}]\right\|^2 \\
& - \langle \nabla f(\boldsymbol{x}_{t,k}),\, \mathbb{E}_{t,k}[\boldsymbol{d}_{t,k}]\rangle 
\end{aligned}
\tag{27}
$$

$$
\begin{aligned}
= & \frac{1}{2a}\left\|\nabla f(\boldsymbol{y}_{t,k}) - \nabla f(\boldsymbol{x}_{t,k})\right\|^2 - \frac{1-a}{2}\left\|\mathbb{E}_{t,k}[\boldsymbol{d}_{t,k}]\right\|^2 \\
& - \frac{1}{2}\left\|\nabla f(\boldsymbol{x}_{t,k})\right\|^2 + \frac{1}{2}\left\|\nabla f(\boldsymbol{x}_{t,k}) - \mathbb{E}_{t,k}[\boldsymbol{d}_{t,k}]\right\|^2
\end{aligned}
\tag{28}
$$

$$
\begin{aligned}
\leq & \frac{L^2}{2a}\left\|\boldsymbol{y}_{t,k} - \boldsymbol{x}_{t,k}\right\|^2 - \frac{1-a}{2}\left\|\mathbb{E}_{t,k}[\boldsymbol{d}_{t,k}]\right\|^2 \\
& - \frac{1}{2}\left\|\nabla f(\boldsymbol{x}_{t,k})\right\|^2 + \frac{1}{2}\left\|\nabla f(\boldsymbol{x}_{t,k}) - \mathbb{E}_{t,k}[\boldsymbol{d}_{t,k}]\right\|^2
\end{aligned}
\tag{29}
$$

where (27) comes from Fact 4 (24), and $a > 0$ is constant. For simplicity, we directly set $a = 0.5$. Eqn. (28) uses Fact 2 $\langle \mathbf{a},\, \mathbf{b}\rangle = \frac{1}{2}\|\mathbf{a}\|^2 + \frac{1}{2}\|\mathbf{b}\|^2 - \frac{1}{2}\|\mathbf{a} - \mathbf{b}\|^2$. Furthermore, according to the definition of $\boldsymbol{y}_{t,k}$, it can be shown that

$$
\|\boldsymbol{y}_{t,k} - \boldsymbol{x}_{t,k}\|^2 = \left\|\left(1 - \frac{\alpha}{1-\beta}\right)\sum_{j=0}^{k-1}\gamma\boldsymbol{d}_{t,j} + \boldsymbol{y}_{t,0} - \boldsymbol{x}_{t,0}\right\|^2
\tag{30}
$$

$$
\leq 2\gamma^2\left(1 - \frac{\alpha}{1-\beta}\right)^2\left\|\sum_{j=0}^{k-1}\boldsymbol{d}_{t,j}\right\|^2 + 2\left\|\boldsymbol{y}_{t,0} - \boldsymbol{x}_{t,0}\right\|^2
\tag{31}
$$

$$
= 2\gamma^2\left(1 - \frac{\alpha}{1-\beta}\right)^2\left\|\sum_{j=0}^{k-1}\boldsymbol{d}_{t,j}\right\|^2 + \frac{2\beta^2}{(1-\beta)^2}\left\|\boldsymbol{x}_{t,0} - \boldsymbol{x}_{t-1,0}\right\|^2.
\tag{32}
$$

Substituting (32) into (29), it follows that

$$
\begin{aligned}
-\langle \nabla f(\boldsymbol{y}_{t,k}),\, \mathbb{E}_{t,k}[\boldsymbol{d}_{t,k}]\rangle \leq & -\frac{1}{2}\left\|\nabla f(\boldsymbol{x}_{t,k})\right\|^2 - \frac{1}{4}\left\|\mathbb{E}_{t,k}[\boldsymbol{d}_{t,k}]\right\|^2 + \frac{1}{2}\left\|\nabla f(\boldsymbol{x}_{t,k}) - \mathbb{E}_{t,k}[\boldsymbol{d}_{t,k}]\right\|^2 \\
& + 2\gamma^2 L^2\left(1 - \frac{\alpha}{1-\beta}\right)^2\left\|\sum_{j=0}^{k-1}\boldsymbol{d}_{t,j}\right\|^2 + \frac{2L^2\beta^2}{(1-\beta)^2}\left\|\boldsymbol{x}_{t,0} - \boldsymbol{x}_{t-1,0}\right\|^2.
\end{aligned}
\tag{33}
$$

Moreover, for the second term in (26), we have

$$
\mathbb{E}_{t,k}\left[\left\|\boldsymbol{d}_{t,k}\right\|^2\right] = \left\|\mathbb{E}_{t,k}[\boldsymbol{d}_{t,k}]\right\|^2 + \mathbb{E}_{t,k}\left[\left\|\boldsymbol{d}_{t,k} - \mathbb{E}_{t,k}[\boldsymbol{d}_{t,k}]\right\|^2\right].
\tag{34}
$$

Then, plugging (33) and (34) into (26),

$$
\begin{aligned}
\mathbb{E}_{t,k}[f(\boldsymbol{y}_{t,k+1})] - f(\boldsymbol{y}_{t,k}) \leq & -\frac{\gamma_{\text{eff}}}{2}\left\|\nabla f(\boldsymbol{x}_{t,k})\right\|^2 - \frac{\gamma_{\text{eff}}}{2}\left(\frac{1}{2} - \gamma_{\text{eff}}L\right)\left\|\mathbb{E}_{t,k}[\boldsymbol{d}_{t,k}]\right\|^2 \\
& + \frac{\gamma_{\text{eff}}^2 L}{2}\mathbb{E}_{t,k}\left[\left\|\boldsymbol{d}_{t,k} - \mathbb{E}_{t,k}[\boldsymbol{d}_{t,k}]\right\|^2\right] + \frac{\gamma_{\text{eff}}}{2}\left\|\nabla f(\boldsymbol{x}_{t,k}) - \mathbb{E}_{t,k}[\boldsymbol{d}_{t,k}]\right\|^2 \\
& + 2\gamma_{\text{eff}}\gamma^2 L^2\left(1 - \frac{\alpha}{1-\beta}\right)^2\left\|\sum_{j=0}^{k-1}\boldsymbol{d}_{t,j}\right\|^2 + \frac{2\gamma_{\text{eff}}\beta^2 L^2}{(1-\beta)^2}\left\|\boldsymbol{x}_{t,0} - \boldsymbol{x}_{t-1,0}\right\|^2
\end{aligned}
\tag{35}
$$

where $\gamma_{\text{eff}} = \alpha\gamma/(1-\beta)$. Taking the total expectation,

$$
\begin{aligned}
\mathbb{E}[f(\boldsymbol{y}_{t,k+1}) - f(\boldsymbol{y}_{t,k})] \leq & -\frac{\gamma_{\text{eff}}}{2}\mathbb{E}\left\|\nabla f(\boldsymbol{x}_{t,k})\right\|^2 - \frac{\gamma_{\text{eff}}}{2}\left(\frac{1}{2} - \gamma_{\text{eff}}L\right)\mathbb{E}\left\|\mathbb{E}_{t,k}[\boldsymbol{d}_{t,k}]\right\|^2 \\
& + \underbrace{\frac{\gamma_{\text{eff}}^2 L}{2}\mathbb{E}\left[\left\|\boldsymbol{d}_{t,k} - \mathbb{E}_{t,k}[\boldsymbol{d}_{t,k}]\right\|^2\right] + \frac{\gamma_{\text{eff}}}{2}\mathbb{E}\left\|\nabla f(\boldsymbol{x}_{t,k}) - \mathbb{E}_{t,k}[\boldsymbol{d}_{t,k}]\right\|^2}_{N_1(t,k)} \\
& + \underbrace{2\gamma_{\text{eff}}\gamma^2 L^2\left(1 - \frac{\alpha}{1-\beta}\right)^2\mathbb{E}\left\|\sum_{j=0}^{k-1}\boldsymbol{d}_{t,j}\right\|^2}_{N_2(t,k)} + \underbrace{\frac{2\gamma_{\text{eff}}\beta^2 L^2}{(1-\beta)^2}\mathbb{E}\left\|\boldsymbol{x}_{t,0} - \boldsymbol{x}_{t-1,0}\right\|^2}_{N_3(t)}.
\end{aligned}
$$

$$(36)$$

Summing from $k = 0$ to $k = \tau - 1$, we have

$$
\mathbb{E}[f(\boldsymbol{y}_{t+1,0}) - f(\boldsymbol{y}_{t,0})] = \mathbb{E}[f(\boldsymbol{y}_{t,\tau}) - f(\boldsymbol{y}_{t,0})] \tag{37}
$$

$$
\begin{aligned}
\leq & -\frac{\gamma_{\text{eff}}}{2}\sum_{k=0}^{\tau-1}\mathbb{E}\left\|\nabla f(\boldsymbol{x}_{t,k})\right\|^2 - \frac{\gamma_{\text{eff}}}{2}\left(\frac{1}{2} - \gamma_{\text{eff}}L\right)\sum_{k=0}^{\tau-1}\mathbb{E}\left\|\mathbb{E}_{t,k}[\boldsymbol{d}_{t,k}]\right\|^2 \\
& + \frac{\gamma_{\text{eff}}^2 L\tau V}{2} + \frac{\gamma_{\text{eff}}}{2}\sum_{k=0}^{\tau-1}\mathbb{E}\left\|\nabla f(\boldsymbol{x}_{t,k}) - \mathbb{E}_{t,k}[\boldsymbol{d}_{t,k}]\right\|^2 + \sum_{k=0}^{\tau-1}N_2(t,k) + \sum_{k=0}^{\tau-1}N_3(t),
\end{aligned}
$$

$$(38)$$

where $N_1(t,k)$, $N_2(t,k)$, and $N_3(t)$ are as defined in (36). Summing from $t = 0$ to $t = T - 1$ and dividing both side by total iterations $K = \tau T$,

$$
\begin{aligned}
\frac{\mathbb{E}[f(\boldsymbol{y}_{T,0}) - f(\boldsymbol{y}_{0,0})]}{K} \leq & -\frac{\gamma_{\text{eff}}}{2K}\sum_{t=0}^{T-1}\sum_{k=0}^{\tau-1}\mathbb{E}\left\|\nabla f(\boldsymbol{x}_{t,k})\right\|^2 - \frac{\gamma_{\text{eff}}}{2K}\left(\frac{1}{2} - \gamma_{\text{eff}}L\right)\sum_{t=0}^{T-1}\sum_{k=0}^{\tau-1}\mathbb{E}\left\|\mathbb{E}_{t,k}[\boldsymbol{d}_{t,k}]\right\|^2 \\
& + \frac{\gamma_{\text{eff}}^2 LV}{2} + \frac{\gamma_{\text{eff}}}{2K}\sum_{t=0}^{T-1}\sum_{k=0}^{\tau-1}\mathbb{E}\left\|\nabla f(\boldsymbol{x}_{t,k}) - \mathbb{E}_{t,k}[\boldsymbol{d}_{t,k}]\right\|^2 \\
& + \frac{1}{K}\sum_{t=0}^{T-1}\sum_{k=0}^{\tau-1}N_2(t,k) + \frac{1}{K}\sum_{t=0}^{T-1}\sum_{k=0}^{\tau-1}N_3(t).
\end{aligned}
$$

$$(39)$$

Now, we are going to further expand the expressions of the last two terms in (39).

### D.3.1 BOUNDING $N_2(t,k)$

Using the fact $\|a + b\|^2 \leq 2\|a\|^2 + 2\|b\|^2$, we have

$$
\left\|\sum_{j=0}^{k-1}\boldsymbol{d}_{t,j}\right\|^2 \leq 2\left\|\sum_{j=0}^{k-1}(\boldsymbol{d}_{t,j} - \mathbb{E}_{t,j}[\boldsymbol{d}_{t,j}])\right\|^2 + 2\left\|\sum_{j=0}^{k-1}\mathbb{E}_{t,j}[\boldsymbol{d}_{t,j}]\right\|^2 \tag{40}
$$

$$
\leq 2\left\|\sum_{j=0}^{k-1}(\boldsymbol{d}_{t,j} - \mathbb{E}_{t,j}[\boldsymbol{d}_{t,j}])\right\|^2 + 2k\sum_{j=0}^{k-1}\|\mathbb{E}_{t,j}[\boldsymbol{d}_{t,j}]\|^2 \tag{41}
$$

where the last inequality comes from Fact 3. Then, taking the total expectation and summing over the $t$-th outer iteration,

$$\mathbb{E}\left[\sum_{k=0}^{\tau-1}\left\|\sum_{j=0}^{k-1}\boldsymbol{d}_{t,j}\right\|^2\right] \leq 2\mathbb{E}\left[\sum_{k=0}^{\tau-1}\left\|\sum_{j=0}^{k-1}(\boldsymbol{d}_{t,j}-\mathbb{E}_{t,j}[\boldsymbol{d}_{t,j}])\right\|^2\right] + 2\sum_{k=0}^{\tau-1}k\sum_{j=0}^{k-1}\mathbb{E}\left\|\mathbb{E}_{t,j}[\boldsymbol{d}_{t,j}]\right\|^2 \quad (42)$$

$$=2\sum_{k=0}^{\tau-1}\sum_{j=0}^{k-1}\mathbb{E}\left[\|\boldsymbol{d}_{t,j}-\mathbb{E}_{t,j}[\boldsymbol{d}_{t,j}]\|^2\right] + 2\sum_{k=0}^{\tau-1}k\sum_{j=0}^{k-1}\mathbb{E}\left\|\mathbb{E}_{t,j}[\boldsymbol{d}_{t,j}]\right\|^2 \quad (43)$$

$$\leq 2V\sum_{k=0}^{\tau-1}k + 2\sum_{k=0}^{\tau-1}k\sum_{j=0}^{\tau-1}\mathbb{E}\left\|\mathbb{E}_{t,j}[\boldsymbol{d}_{t,j}]\right\|^2 \quad (44)$$

$$=\tau(\tau-1)V + \tau(\tau-1)\sum_{k=0}^{\tau-1}\mathbb{E}\left\|\mathbb{E}_{t,k}[\boldsymbol{d}_{t,k}]\right\|^2 \quad (45)$$

where (43) uses the following fact:

$$\mathbb{E}\left[\left\|\sum_{j=0}^{k-1}(\boldsymbol{d}_{t,j}-\mathbb{E}_{t,j}[\boldsymbol{d}_{t,j}])\right\|^2\right] = \sum_{j=0}^{k-1}\mathbb{E}\left[\|\boldsymbol{d}_{t,j}-\mathbb{E}_{t,j}[\boldsymbol{d}_{t,j}]\|^2\right]$$

$$+ 2\sum_{j=0}^{k-1}\sum_{l=j+1}^{k-1}\mathbb{E}\left\langle\boldsymbol{d}_{t,j}-\mathbb{E}_{t,j}[\boldsymbol{d}_{t,j}], \boldsymbol{d}_{t,l}-\mathbb{E}_{t,l}[\boldsymbol{d}_{t,l}]\right\rangle \quad (46)$$

$$= \sum_{j=0}^{k-1}\mathbb{E}\left[\|\boldsymbol{d}_{t,j}-\mathbb{E}_{t,j}[\boldsymbol{d}_{t,j}]\|^2\right]$$

$$+ 2\sum_{j=0}^{k-1}\sum_{l=j+1}^{k-1}\mathbb{E}\left\langle\boldsymbol{d}_{t,j}-\mathbb{E}_{t,j}[\boldsymbol{d}_{t,j}], \mathbb{E}_{t,l}[\boldsymbol{d}_{t,l}-\mathbb{E}_{t,l}[\boldsymbol{d}_{t,l}]]\right\rangle \quad (47)$$

$$= \sum_{j=0}^{k-1}\mathbb{E}\left[\|\boldsymbol{d}_{t,j}-\mathbb{E}_{t,j}[\boldsymbol{d}_{t,j}]\|^2\right]. \quad (48)$$

As a result, we end up with the following

$$\sum_{k=0}^{\tau-1}N_2(t,k) \leq 2\gamma_{\text{eff}}^3 L^2\tau(\tau-1)\frac{(1-\beta-\alpha)^2}{\alpha^2}\left[V+\sum_{k=0}^{\tau-1}\mathbb{E}\left\|\mathbb{E}_{t,k}[\boldsymbol{d}_{t,k}]\right\|^2\right], \quad (49)$$

$$\sum_{t=0}^{T-1}\sum_{k=0}^{\tau-1}N_2(t,k) \leq 2\gamma_{\text{eff}}^3 L^2\tau(\tau-1)\frac{(1-\beta-\alpha)^2}{\alpha^2}\left[TV+\sum_{t=0}^{T-1}\sum_{k=0}^{\tau-1}\mathbb{E}\left\|\mathbb{E}_{t,k}[\boldsymbol{d}_{t,k}]\right\|^2\right] \quad (50)$$

$$\frac{1}{K}\sum_{t=0}^{T-1}\sum_{k=0}^{\tau-1}N_2(t,k) \leq 2\gamma_{\text{eff}}^3 L^2\tau(\tau-1)\frac{(1-\beta-\alpha)^2}{\alpha^2}\left[\frac{V}{\tau}+\frac{1}{K}\sum_{t=0}^{T-1}\sum_{k=0}^{\tau-1}\mathbb{E}\left\|\mathbb{E}_{t,k}[\boldsymbol{d}_{t,k}]\right\|^2\right] \quad (51)$$

where $K=\tau T$ denotes the total steps.

### D.3.2 BOUNDING $N_3(t)$

From the update rule (2), (3) and (18), we have

$$\|\boldsymbol{x}_{t,0} - \boldsymbol{x}_{t-1,0}\|^2 = \alpha^2\gamma^2 \left\|\sum_{k=0}^{\tau-1} \boldsymbol{d}_{t-1,k} + \beta\boldsymbol{u}_{t-1}\right\|^2 \tag{52}$$

$$= \alpha^2\gamma^2 \left\|\sum_{k=0}^{\tau-1} \boldsymbol{d}_{t-1,k} + \beta\sum_{k=0}^{\tau-1} \boldsymbol{d}_{t-2,k} + \beta^2\boldsymbol{u}_{t-2}\right\|^2 \tag{53}$$

$$= \alpha^2\gamma^2 \left\|\sum_{s=0}^{t-1} \beta^{t-1-s} \left(\sum_{k=0}^{\tau-1} \boldsymbol{d}_{s,k}\right)\right\|^2 \tag{54}$$

$$\leq 2\alpha^2\gamma^2 \underbrace{\left\|\sum_{s=0}^{t-1} \beta^{t-1-s} \left(\sum_{k=0}^{\tau-1}(\boldsymbol{d}_{s,k} - \mathbb{E}_{s,k}[\boldsymbol{d}_{s,k}])\right)\right\|^2}_{T_1}$$

$$+ 2\alpha^2\gamma^2 \underbrace{\left\|\sum_{s=0}^{t-1} \beta^{t-1-s} \left(\sum_{k=0}^{\tau-1} \mathbb{E}_{s,k}[\boldsymbol{d}_{s,k}]\right)\right\|^2}_{T_2} \tag{55}$$

For the first term $T_1$, taking the total expectation, we get

$$\mathbb{E}[T_1] = \mathbb{E}\left[\left\|\sum_{s=0}^{t-1} \beta^{t-1-s} \left(\sum_{k=0}^{\tau-1}(\boldsymbol{d}_{s,k} - \mathbb{E}_{s,k}[\boldsymbol{d}_{s,k}])\right)\right\|^2\right] \tag{56}$$

$$= \sum_{s=0}^{t-1} \beta^{2(t-1-s)} \mathbb{E}\left[\left\|\sum_{k=0}^{\tau-1}(\boldsymbol{d}_{s,k} - \mathbb{E}_{s,k}[\boldsymbol{d}_{s,k}])\right\|^2\right] \tag{57}$$

$$= \sum_{s=0}^{t-1} \beta^{2(t-1-s)} \sum_{k=0}^{\tau-1} \mathbb{E}\left[\|\boldsymbol{d}_{s,k} - \mathbb{E}_{s,k}[\boldsymbol{d}_{s,k}])\|^2\right] \tag{58}$$

$$\leq \tau V \sum_{s=0}^{t-1} \beta^{2(t-1-s)} = \tau V(1 + \beta^2 + \beta^4 + \cdots + \beta^{2(t-1)}) \leq \frac{\tau V}{1-\beta^2} \tag{59}$$

where (57) and (58) are derived using the same routine as (46) to (48). Similarly, for the second term $T_2$ in (55), according to Fact 4,

$$\mathbb{E}[T_2] \leq \left(\sum_{s=0}^{t-1} \beta^{t-1-s}\right) \sum_{s=0}^{t-1} \beta^{t-1-s} \mathbb{E}\left[\left\|\sum_{k=0}^{\tau-1} \mathbb{E}_{s,k} \boldsymbol{d}_{s,k}\right\|^2\right] \tag{60}$$

$$\leq \tau \left(\sum_{s=0}^{t-1} \beta^{t-1-s}\right) \sum_{s=0}^{t-1} \beta^{t-1-s} \sum_{k=0}^{\tau-1} \mathbb{E}\left[\|\mathbb{E}_{s,k} \boldsymbol{d}_{s,k}\|^2\right] \tag{61}$$

$$\leq \frac{\tau}{1-\beta} \sum_{s=0}^{t-1} \beta^{t-1-s} \sum_{k=0}^{\tau-1} \mathbb{E}\left[\|\mathbb{E}_{s,k} \boldsymbol{d}_{s,k}\|^2\right]. \tag{62}$$

Substituting (59) and (62) back into (55) and summing over the $t$-th outer iteration, we have

$$\sum_{k=0}^{\tau-1} N_3(t) \leq \frac{4\gamma_{\text{eff}}^3 L^2 \beta^2 \tau^2 V}{(1-\beta^2)} + \frac{4\gamma_{\text{eff}}^3 L^2 \beta^2 \tau^2}{(1-\beta)} \sum_{s=0}^{t-1} \beta^{t-1-s} \sum_{k=0}^{\tau-1} \mathbb{E}\left[\left\|\mathbb{E}_{s,k} \boldsymbol{d}_{s,k}\right\|^2\right], \tag{63}$$

$$\sum_{t=0}^{T-1}\sum_{k=0}^{\tau-1} N_3(t) \leq \frac{4K\gamma_{\text{eff}}^3 L^2 \beta^2 \tau V}{(1-\beta^2)} + \frac{4\gamma_{\text{eff}}^3 L^2 \beta^2 \tau^2}{(1-\beta)} \sum_{t=0}^{T-1}\sum_{s=0}^{t-1} \beta^{t-1-s} \sum_{k=0}^{\tau-1} \mathbb{E}\left[\left\|\mathbb{E}_{s,k} \boldsymbol{d}_{s,k}\right\|^2\right] \tag{64}$$

$$= \frac{4K\gamma_{\text{eff}}^3 L^2 \beta^2 \tau V}{(1-\beta^2)} + \frac{4\gamma_{\text{eff}}^3 L^2 \beta^2 \tau^2}{(1-\beta)} \sum_{t=0}^{T-2}\sum_{k=0}^{\tau-1} \mathbb{E}\left[\left\|\mathbb{E}_{t,k} \boldsymbol{d}_{t,k}\right\|^2\right] \sum_{s=t+1}^{T-1} \beta^{T-1-s} \tag{65}$$

$$\leq \frac{4K\gamma_{\text{eff}}^3 L^2 \beta^2 \tau V}{(1-\beta^2)} + \frac{4\gamma_{\text{eff}}^3 L^2 \beta^2 \tau^2}{(1-\beta)^2} \sum_{t=0}^{T-2}\sum_{k=0}^{\tau-1} \mathbb{E}\left[\left\|\mathbb{E}_{t,k} \boldsymbol{d}_{t,k}\right\|^2\right] \tag{66}$$

$$\leq \frac{4K\gamma_{\text{eff}}^3 L^2 \beta^2 \tau V}{(1-\beta^2)} + \frac{4\gamma_{\text{eff}}^3 L^2 \beta^2 \tau^2}{(1-\beta)^2} \sum_{t=0}^{T-1}\sum_{k=0}^{\tau-1} \mathbb{E}\left[\left\|\mathbb{E}_{t,k} \boldsymbol{d}_{t,k}\right\|^2\right], \tag{67}$$

$$\frac{1}{K}\sum_{t=0}^{T-1}\sum_{k=0}^{\tau-1} N_3(t) \leq \frac{4\gamma_{\text{eff}}^3 L^2 \beta^2 \tau V}{(1-\beta^2)} + \frac{4\gamma_{\text{eff}}^3 L^2 \beta^2 \tau^2}{(1-\beta)^2} \frac{1}{K}\sum_{t=0}^{T-1}\sum_{k=0}^{\tau-1} \mathbb{E}\left[\left\|\mathbb{E}_{t,k} \boldsymbol{d}_{t,k}\right\|^2\right]. \tag{68}$$

### D.3.3 FINAL RESULTS

Plugging (51) and (68) back into (39), one can obtain

$$
\begin{aligned}
\frac{\mathbb{E}[f(\boldsymbol{y}_{T,0}) - f(\boldsymbol{y}_{0,0})]}{K} \leq & -\frac{\gamma_{\text{eff}}}{2K}\sum_{t=0}^{T-1}\sum_{k=0}^{\tau-1} \mathbb{E}\left\|\nabla f(\boldsymbol{x}_{t,k})\right\|^2 - \frac{\gamma_{\text{eff}}}{2K}C_1 \sum_{t=0}^{T-1}\sum_{k=0}^{\tau-1} \mathbb{E}\left\|\mathbb{E}_{t,k}[\boldsymbol{d}_{t,k}]\right\|^2 \\
& + \frac{\gamma_{\text{eff}}^2 L V}{2}\left[1 + 4\gamma_{\text{eff}}L(\tau-1)\left(\frac{1-\beta}{\alpha} - 1\right)^2 + \frac{8\gamma_{\text{eff}}L\tau\beta^2}{(1-\beta^2)}\right] \\
& + \frac{\gamma_{\text{eff}}}{2K}\sum_{t=0}^{T-1}\sum_{k=0}^{\tau-1} \mathbb{E}\left\|\nabla f(\boldsymbol{x}_{t,k}) - \mathbb{E}_{t,k}[\boldsymbol{d}_{t,k}]\right\|^2
\end{aligned}
\tag{69}
$$

where $C_1 = 1/2 - \gamma_{\text{eff}}L - 4\gamma_{\text{eff}}^2 L^2 \tau(\tau-1)(1-\beta-\alpha)^2/\alpha^2 - 8\gamma_{\text{eff}}^2 L^2 \tau^2 \beta^2/(1-\beta)^2$. When the constants satisfy

$$\frac{1}{2} - \gamma_{\text{eff}}L - 4\gamma_{\text{eff}}^2 L^2 \tau(\tau-1)\frac{(1-\beta-\alpha)^2}{\alpha^2} - \frac{8\gamma_{\text{eff}}^2 L^2 \tau^2 \beta^2}{(1-\beta)^2} \geq 0, \tag{70}$$

we have

$$
\begin{aligned}
\frac{\mathbb{E}[f(\boldsymbol{y}_{T,0}) - f(\boldsymbol{y}_{0,0})]}{K} \leq & -\frac{\gamma_{\text{eff}}}{2K}\sum_{t=0}^{T-1}\sum_{k=0}^{\tau-1} \mathbb{E}\left\|\nabla f(\boldsymbol{x}_{t,k})\right\|^2 + \frac{\gamma_{\text{eff}}}{2K}\sum_{t=0}^{T-1}\sum_{k=0}^{\tau-1} \mathbb{E}\left\|\nabla f(\boldsymbol{x}_{t,k}) - \mathbb{E}_{t,k}[\boldsymbol{d}_{t,k}]\right\|^2 \\
& + \frac{\gamma_{\text{eff}}^2 L V}{2}\left[1 + 4\gamma_{\text{eff}}L(\tau-1)\left(\frac{1-\beta}{\alpha} - 1\right)^2 + \frac{8\gamma_{\text{eff}}L\tau\beta^2}{(1-\beta^2)}\right].
\end{aligned}
\tag{71}
$$

After rearranging terms, we get

$$
\begin{aligned}
\frac{1}{K}\sum_{t=0}^{T-1}\sum_{k=0}^{\tau-1} \mathbb{E}\left\|\nabla f(\boldsymbol{x}_{t,k})\right\|^2 \leq & \frac{2\mathbb{E}[f(\boldsymbol{y}_{0,0}) - f(\boldsymbol{y}_{T,0})]}{\gamma_{\text{eff}}K} + \frac{1}{K}\sum_{t=0}^{T-1}\sum_{k=0}^{\tau-1} \mathbb{E}\left\|\nabla f(\boldsymbol{x}_{t,k}) - \mathbb{E}_{t,k}[\boldsymbol{d}_{t,k}]\right\|^2 \\
& + \gamma_{\text{eff}} L V\left[1 + 4\gamma_{\text{eff}}L(\tau-1)\left(\frac{1-\beta}{\alpha} - 1\right)^2 + \frac{8\gamma_{\text{eff}}L\tau\beta^2}{1-\beta^2}\right].
\end{aligned}
\tag{72}
$$

Furthermore, since $\boldsymbol{y}_{0,0} = \boldsymbol{x}_{0,0} - \beta \boldsymbol{x}_{-1,0}/(1-\beta) = \boldsymbol{x}_{0,0}$ and $f(\boldsymbol{y}_{T,0}) \geq f_{\text{inf}}$, the above upper bound can be simplified as

$$
\frac{1}{K} \sum_{t=0}^{T-1} \sum_{k=0}^{\tau-1} \mathbb{E} \|\nabla f(\boldsymbol{x}_{t,k})\|^2 \leq \frac{2\left(f(\boldsymbol{x}_{0,0}) - f_{\text{inf}}\right)}{\gamma_{\text{eff}} K} + \frac{1}{K} \sum_{t=0}^{T-1} \sum_{k=0}^{\tau-1} \mathbb{E} \|\nabla f(\boldsymbol{x}_{t,k}) - \mathbb{E}_{t,k}[\boldsymbol{d}_{t,k}]\|^2
$$
$$
+ \gamma_{\text{eff}} L V \left[1 + 4\gamma_{\text{eff}} L(\tau-1) \left(\frac{1-\beta}{\alpha} - 1\right)^2 + \frac{8\gamma_{\text{eff}} L \tau \beta^2}{1-\beta^2}\right]. \quad (73)
$$

If we set $\gamma_{\text{eff}} = \sqrt{\frac{m}{K}}$, then

$$
\frac{1}{K} \sum_{t=0}^{T-1} \sum_{k=0}^{\tau-1} \mathbb{E} \|\nabla f(\boldsymbol{x}_{t,k})\|^2 \leq \frac{2\left(f(\boldsymbol{x}_{0,0}) - f_{\text{inf}}\right)}{\sqrt{mK}} + \frac{1}{K} \sum_{t=0}^{T-1} \sum_{k=0}^{\tau-1} \mathbb{E} \|\nabla f(\boldsymbol{x}_{t,k}) - \mathbb{E}_{t,k}[\boldsymbol{d}_{t,k}]\|^2
$$
$$
+ \frac{mLV}{\sqrt{mK}} + \frac{4mL^2(\tau-1)}{K} \left(\frac{1-\beta}{\alpha} - 1\right)^2 + \frac{8mL^2 \tau \beta^2}{K(1-\beta^2)}. \quad (74)
$$

Recall the learning rate constraint is

$$
\frac{1}{2} - \gamma_{\text{eff}} L - 4\gamma_{\text{eff}}^2 L^2 \tau(\tau-1) \frac{(1-\beta-\alpha)^2}{\alpha^2} - 8\gamma_{\text{eff}}^2 L^2 \tau^2 \frac{\beta^2}{(1-\beta)^2} \geq 0. \quad (75)
$$

When $\gamma_{\text{eff}} = \frac{\sqrt{m}}{\sqrt{K}}$, the constraint can be rewritten as

$$
\frac{1}{2} \geq \sqrt{\frac{mL^2}{K}} + 4\tau(\tau-1)\frac{(1-\beta-\alpha)^2}{\alpha^2}\frac{mL^2}{K} + 8\tau^2 \frac{\beta^2}{(1-\beta)^2}\frac{mL^2}{K}. \quad (76)
$$

After rearranging, we have

$$
\frac{K}{mL^2} - 2\sqrt{\frac{K}{mL^2}} + 1 \geq 8\tau(\tau-1)\frac{(1-\beta-\alpha)^2}{\alpha^2} + 16\tau^2 \frac{\beta^2}{(1-\beta)^2} + 1, \quad (77)
$$

$$
\frac{K}{mL^2} - 1 \geq \left(8\tau(\tau-1)\frac{(1-\beta-\alpha)^2}{\alpha^2} + 16\tau^2 \frac{\beta^2}{(1-\beta)^2} + 1\right)^{\frac{1}{2}}, \quad (78)
$$

$$
\frac{K}{mL^2} \geq 1 + \left(8\tau(\tau-1)\frac{(1-\beta-\alpha)^2}{\alpha^2} + 16\tau^2 \frac{\beta^2}{(1-\beta)^2} + 1\right)^{\frac{1}{2}}. \quad (79)
$$

Furthermore, note that

$$
\left(8\tau(\tau-1)\frac{(1-\beta-\alpha)^2}{\alpha^2} + 16\tau^2 \frac{\beta^2}{(1-\beta)^2} + 1\right)^{\frac{1}{2}}
$$
$$
\leq \left(9\tau^2 \frac{(1-\beta-\alpha)^2}{\alpha^2} + 16\tau^2 \frac{\beta^2}{(1-\beta)^2} + 1\right)^{\frac{1}{2}} \quad (80)
$$
$$
\leq \sqrt{3} \max\left\{\frac{3\tau(1-\beta-\alpha)}{\alpha}, \frac{4\tau\beta}{1-\beta}, 1\right\}. \quad (81)
$$

Therefore, when $K \geq mL^2 \left(1 + \sqrt{3} \max\left\{\frac{3\tau(1-\beta-\alpha)}{\alpha}, \frac{4\tau\beta}{1-\beta}, 1\right\}\right)$, the condition (79) must be satisfied.

### D.4 SPECIAL CASE 1: BLOCKWISE MODEL UPDATE FILTERING (BMUF)

In this case, the inner optimizer is Local-SGD. That is,

$$
\boldsymbol{d}_{t,k} = \frac{1}{m} \sum_{i=1}^{m} \nabla F(\boldsymbol{x}_{t,k}^{(i)}; \xi_{t,k}^{(i)}), \ \mathbb{E}_{t,k}[\boldsymbol{d}_{t,k}] = \frac{1}{m} \sum_{i=1}^{m} \nabla f(\boldsymbol{x}_{t,k}^{(i)}). \quad (82)
$$

Since all worker nodes are averaged after every $\tau$ iterations, we have $\boldsymbol{x}_{t,0}^{(i)} = \boldsymbol{x}_{t,0}, \forall i$. Besides, it is easy to validate that $V = \sigma^2/m$.

According to previous literature on the convergence of Local-SGD (Wang & Joshi, 2018; Yu et al., 2019a), we can directly get the following upper bound.

$$A = \frac{1}{K} \sum_{t=0}^{T-1} \sum_{k=0}^{\tau-1} \mathbb{E} \left\| \nabla f(\boldsymbol{x}_{t,k}) - \mathbb{E}_{t,k}[\boldsymbol{d}_{t,k}] \right\|^2 \tag{83}$$

$$= \frac{1}{K} \sum_{t=0}^{T-1} \sum_{k=0}^{\tau-1} \mathbb{E} \left\| \nabla f(\boldsymbol{x}_{t,k}) - \frac{1}{m} \sum_{i=1}^{m} \nabla f(\boldsymbol{x}_{t,k}^{(i)}) \right\|^2 \tag{84}$$

$$\leq \frac{1}{mK} \sum_{t=0}^{T-1} \sum_{k=0}^{\tau-1} \sum_{i=1}^{m} \mathbb{E} \left\| \nabla f(\boldsymbol{x}_{t,k}) - \nabla f(\boldsymbol{x}_{t,k}^{(i)}) \right\|^2 \tag{85}$$

$$\leq \frac{L^2}{mK} \sum_{t=0}^{T-1} \sum_{k=0}^{\tau-1} \sum_{i=1}^{m} \mathbb{E} \left\| \boldsymbol{x}_{t,k} - \boldsymbol{x}_{t,k}^{(i)} \right\|^2 \tag{86}$$

$$\leq \frac{2\gamma^2 L^2 \sigma^2 \tau}{1 - 12\gamma^2 L^2 \tau^2} + \frac{6\gamma^2 L^2 \zeta^2 \tau^2}{1 - 12\gamma^2 L^2 \tau^2}. \tag{87}$$

When $\gamma L \tau \leq \frac{1}{6}$, we have $1/(1 - 12\gamma^2 L^2 \tau^2) \geq 3/2$. It follows that

$$\frac{1}{K} \sum_{t=0}^{T-1} \sum_{k=0}^{\tau-1} \mathbb{E} \left\| \nabla f(\boldsymbol{x}_{t,k}) - \mathbb{E}_{t,k}[\boldsymbol{d}_{t,k}] \right\|^2 \leq 3\gamma^2 L^2 \sigma^2 \tau + 9\gamma^2 L^2 \zeta^2 \tau^2. \tag{88}$$

Substituting (88) into (73) and setting $\frac{\alpha}{1-\beta} \gamma L = \sqrt{m/K}$, we have

$$\frac{1}{K} \sum_{t=0}^{T-1} \sum_{k=0}^{\tau-1} \mathbb{E} \left\| \nabla f(\boldsymbol{x}_{t,k}) \right\|^2 \leq \frac{2L \left( f(\boldsymbol{x}_{0,0}) - f_{\inf} \right) + \sigma^2}{\sqrt{mK}} + \frac{\alpha^2 m}{(1-\beta)^2} \frac{3\sigma^2 \tau + 9\zeta^2 \tau^2}{K} + $$

$$\left( \frac{1-\beta}{\alpha} - 1 \right)^2 \frac{4\sigma^2(\tau - 1)}{K} + \frac{\beta^2}{(1-\beta^2)} \frac{8\sigma^2 \tau}{K} \tag{89}$$

$$= \mathcal{O} \left( \frac{1}{\sqrt{mK}} \right) + \mathcal{O} \left( \frac{m}{K} \right). \tag{90}$$

## D.5 SPECIAL CASE 2: LOOKAHEAD

In this case, the inner optimizer is SGD and $m = 1$. Thus, we have $\beta = 0$, $\mathbb{E}_{t,k}[\boldsymbol{d}_{t,k}] = \nabla f(\boldsymbol{x}_{t,k})$, and $V = \sigma^2$. Therefore,

$$\frac{1}{K} \sum_{t=0}^{T-1} \sum_{k=0}^{\tau-1} \mathbb{E} \left\| \nabla f(\boldsymbol{x}_{t,k}) \right\|^2 \leq \frac{2 \left( f(\boldsymbol{x}_{0,0}) - f_{\inf} \right)}{\alpha \gamma K} + \alpha \gamma L \sigma^2 + 4(1-\alpha)^2 \gamma^2 L^2 (\tau - 1) \sigma^2 \tag{91}$$

It can be observed that when $\alpha = 1$ or $\tau = 1$, the above upper bound reduces to the case of vanilla mini-batch SGD. If we set $\alpha \gamma L = \sqrt{1/K}$, then we have

$$\frac{1}{K} \sum_{t=0}^{T-1} \sum_{k=0}^{\tau-1} \mathbb{E} \left\| \nabla f(\boldsymbol{x}_{t,k}) \right\|^2 \leq \frac{2L \left( f(\boldsymbol{x}_{0,0}) - f_{\inf} \right) + \sigma^2}{\sqrt{K}} + \frac{4(1-\alpha)^2(\tau - 1)\sigma^2}{\alpha^2 K} \tag{92}$$

$$= \mathcal{O} \left( \frac{1}{\sqrt{K}} \right) + \mathcal{O} \left( \frac{1}{K} \right). \tag{93}$$

If the total iterations $K$ is sufficiently large, then the first term will dominate the convergence rate.

