# OpenReview forum: "SlowMo: Improving Communication-Efficient Distributed SGD with Slow Momentum"
_ICLR.cc/2020/Conference — Accept (Poster)_

### Official Review · AnonReviewer2 · 2019-10-23
**Official Blind Review #2**

**Rating:** 6

**Review:**

The authors verify the effect of BMUF[1], which is called slow momentum in this paper, on computer vision and natural language processing tasks with different kinds of local optimizers.  They also provided the theoretical convergence guarantee of BMUF.

The literature survey of this paper is quite good and the experimental results are convincing. However, they should modify their claim that "BMUF is a special case of SlowMomentum".
In classical block momentum version of BMUF, the  update formula is:
u_{t+1} = \beta u_{t} + \alpha (x_{t,0}-x_{t,\tau})
x_{t+1,0} = x_{t,0} - u_{t+1}
\beta is called block momentum and \alpha is block learning rate

in this paper, the update formula becomes:
u_{t+1} = \beta u_{t} +  (x_{t,0}-x_{t,\tau})/\gamma_{t}
x_{t+1,0} = x_{t,0} - \alpha\gamma_{t}u_{t+1}

Obviously this two formula are equivalent. BMUF is a general framework, which can work with different kinds of local optimizer. The author should not narrow down the definition of BMUF as BMUF with SGD as local optimizer and \alpha=1. Actually, \alpha=1 is used in all experiments of this paper.

In conclusion, the organization and writing of this paper is satisfactory, the experiments and theoretical proof is valuable for respective researchers. They should clarify that SlowMomentum is same with BMUF with classical block momentum. I will give a weak accept to this paper.

 [1] Kai Chen and Qiang Huo. Scalable training of deep learning machines by incremental block training with intra-block parallel optimization and blockwise model-update filtering. In 2016 IEEE International Conference on Acoustics, Speech and Signal Processing (ICASSP), pp. 5880–5884, 2016.

**Experience Assessment:**

I have published one or two papers in this area.

**Review Assessment: Checking Correctness Of Derivations And Theory:**

I assessed the sensibility of the derivations and theory.

**Review Assessment: Checking Correctness Of Experiments:**

I carefully checked the experiments.

**Review Assessment: Thoroughness In Paper Reading:**

I read the paper thoroughly.

---

> ### Author Response · Authors · 2019-11-09
> **Response: Thank you for your review**
>
> Thank you very much for the feedback and suggestions. Thanks for pointing out our mischaracterization of BMUF. After re-reading the paper we agree that it mentions other methods than SGD could be used as the local optimizer. In our view, the main way in which SlowMo generalizes BMUF is by allowing workers to communicate during steps of the inner loop; when we use SlowMo with (O)SGP, workers gossip for $\tau$ local iterations and then perform an AllReduce to exactly synchronize before doing the block/slow momentum step. In our experiments on both image classification and machine translations tasks we found that using (O)SGP in the inner loop provided improvements over having workers perform local/independent updates. This also why we chose to refer to the inner loop steps as coming from a “base” optimizer instead of a “local” one. We can do a better job of getting this difference across clearly. We are revising the paper (update will be posted soon) to clarify these points and to make sure we properly describe the contribution of BMUF.

---

### Official Review · AnonReviewer3 · 2019-10-24
**Official Blind Review #3**

**Rating:** 6

**Review:**

Paper summary:
The paper proposes a general framework to improve the optimization and generalization performance of several communication efficient algorithms, including local SGD, SGP. A convergence analysis is provided for smooth non-convex losses.

Score: Accept.

Detailed comments:
Pros:
* The paper is written in a clear and well-organized form. The experimental setup description, as well as the ablation study, provide a clear guideline to use this framework in practice.
* The extensive empirical experiments in this paper justify the effectiveness of the proposed methods.

Cons:
* Providing the convergence analysis is encouraged but more understanding is required. The analysis is quite standard and the convergence rate with extra effect (increasing the upper bound) in Eq.5 cannot explain why it convergences faster and generalizes better (e.g. in Figure 2) than AR-SGD.
* As one main contribution of this paper is in terms of the theoretical convergence guarantees, the related work should precisely mention the recent progress in this area and (maybe) point out the difference compared to the prior work.

Minor comments:
1. It is confusing to talk about the \tau in the main paper and the same \tau notation in Algorithm 3 (default parameter in OSGP). Are these two factors the same?


**Experience Assessment:**

I have read many papers in this area.

**Review Assessment: Checking Correctness Of Derivations And Theory:**

I assessed the sensibility of the derivations and theory.

**Review Assessment: Checking Correctness Of Experiments:**

I assessed the sensibility of the experiments.

**Review Assessment: Thoroughness In Paper Reading:**

I read the paper at least twice and used my best judgement in assessing the paper.

---

> ### Author Response · Authors · 2019-11-09
> **Response: Thank you for your review**
>
> Thank you very much for the feedback and suggestions. We agree that it would be desirable to provide theoretical results justifying why SlowMo leads to accelerated convergence rates, compared to other decentralized methods. To the best of our knowledge, there are still relatively few theoretical results for stochastic momentum methods illustrating when they achieve accelerated rates, even in the centralized/serial setting.
>
> In the initial submission, to stay within the recommended page length, we focused our discussion in Sections 1 and 3 on prior work studying stochastic decentralized/communication-efficient optimization methods for non-convex problems (especially aimed at training deep neural networks), and especially those incorporating some form of momentum. We appreciate that there is a vast literature studying decentralized methods, especially for convex objectives, and there are also exciting recent advances in the theoretical understanding of (centralized) stochastic momentum methods. We have added two paragraphs in Section 3 discussing work along these two lines (revised version will be uploaded soon), including citations to the following papers:
>
> Kevin Scaman,  Francis Bach,  Sebastien Bubeck,  Yin Tat Lee,  and Laurent Massoulie.   Optimal convergence rates for convex distributed optimization in networks. Journal of Machine Learning Research, pp. 1–31, 2019.
>
> Dominic Richards and Patrick Rebeschini. Optimal statistical rates for decentralised non-parametric regression with linear speed-up. In Advances in Neural Information Processing Systems, 2019.
>
> Bugra Can, Mert Gurbuzbalaban, and Lingjiong Zhu.  Accelerated linear convergence of stochastic momentum methods in Wasserstein distances. arXiv preprint arxiv:1901.07445, 2019.
>
> Igor Gitman, Hunter Lang, Pengchuan Zhang, and Lin Xiao.  Understanding the role of momentum in stochastic gradient methods. In Advances in Neural Information Processing Systems, 2019.
>
> Nicolas Loizou and Peter Richtarik.  Momentum and stochastic momentum for stochastic gradient, Newton, proximal point and subspace descent methods. arXiv preprint arxiv:1712.09677, 2017.
>
>
> If you are aware of other references we have missed, please do let us know so we can be sure to put our contributions in context. Thank you also for pointing out the potential confusion around $\tau$ in the appendix; we are modifying it accordingly to fix this issue.

---

### Official Review · AnonReviewer4 · 2019-11-02
**Official Blind Review #4**

**Rating:** 6

**Review:**

The paper presents a simple momentum scheme which can be applied to distributed and decentralized SGD schemes. The scheme proposes to do a sequence inner/local steps of any optimizer without any momentum, but then only apply momentum on the outer level, after each global synchronization round.

The paper is clearly written and experiments are well set up.

In a more general view, I'd encourage the authors to discuss in the paper that momentum can be applied both in the inner and the outer loop. For example both [K19] and [Assran et al. (2019)] have already applied momentum in a similar setting but focusing on the inner level (which would be harder to analyze in theory, but might be a good method in practice).

The provided theoretical convergence result is valuable as a contribution, even if it does not show benefits of momentum over SGD, but at least it shows a slowdown which is bounded, as in single-machine SGD. The authors have done non-trivial work to extend it to the case of communication and local updates, but I didn't have time to check the entire long proof in the appendix.

The experimental results are convincing in comparison to the baselines without momentum.
I would have hoped the authors also add a more clear picture of how the proposed 'outer' momentum would compare to practical 'inner' momentum schemes such as used by [Assran et al. (2019)] and [K19].

The main question for me is on the significance of the contribution. The benefits of momentum are well known in practice in the single-machine case, and theoretically not well understood. The paper here translates this type of results also to the decentralized case. As momentum is common it is not very surprising that it is also beneficial here. The paper could be strengthened by adding comparisons of different inner/outer momentum variants which are unique to the distributed setting.

Minor comments:
- At the end of Section 3, it is said that "Local SGD, ... perform averaging on the model parameters rather than on gradients.".
While not totally wrong, I think one should say that local SGD can easily do averaging of the model changes/deltas, instead of the models themselves.

- Start of Page 3 and of Section 5: potentially clarify notion of 'descent direction', as with SGD those are not technically descent on the original (local or global) objective, but only on the currently sampled stochastic f_i .

UPDATE AFTER REBUTTAL
The discussion and other reviews were helpful, for instance that the paper should still clarify better the fact that the proposed method is just a very minor generalization of BMUF, to some more SGD variants (including decentralized). Nevertheless, reviewers seem to agree that the paper is well-presented in giving a more clear review of such methods, and valuable in improving the theoretical understanding of such distributed momentum methods. I keep the current score.

References:
[K19] Koloskova, Anastasia, et al. "Decentralized Deep Learning with Arbitrary Communication Compression." arXiv preprint arXiv:1907.09356 (2019).

**Experience Assessment:**

I have published one or two papers in this area.

**Review Assessment: Checking Correctness Of Derivations And Theory:**

I assessed the sensibility of the derivations and theory.

**Review Assessment: Checking Correctness Of Experiments:**

I assessed the sensibility of the experiments.

**Review Assessment: Thoroughness In Paper Reading:**

I read the paper at least twice and used my best judgement in assessing the paper.

---

> ### Author Response · Authors · 2019-11-09
> **Response: Thank you for your review**
>
> Thank you very much for the feedback and suggestions. In fact, in all of our experiments the local updates (with or without SlowMo) are all either based on SGD with momentum (on CIFAR-10 and ImageNet) or Adam (on WMT’16 En-De). So when using SlowMo there may actually be two forms of momentum being used. We did run experiments on CIFAR-10 where the local update rule (base optimizer) was plain SGD without momentum. When the local updates do not use momentum, we also observed that introducing SlowMo led to improvements in optimization and generalization. However, using local momentum is better than not using local momentum, and using SlowMo on top of local momentum provides further improvements (as illustrated by the results included in the paper). We realize that we may not have been clear on this point before, and we will update our submission to emphasize that $d^{(i)}_{t,k}$ in line 4 of Alg 1 could be a stochastic gradient $\nabla F_i(x^{(i)}_{t,k}, \xi^{(i)}_{t,k})$, or it could indeed correspond to a local momentum step, e.g., $d^{(i)}_{t,k} = \beta^{(i)}_{\text{local}} d^{(i)}_{t,k-1} + \nabla F_i(x^{(i)}_{t,k}, \xi^{(i)}_{t,k})$. Note that whether or not the local/base optimizer uses momentum is orthogonal to whether one has the workers communicate during the inner loop (e.g., using SGP/OSGP) or not (LocalSGD and LocalAdam). We are also modifying the paper to address the other comments you provided, including a citation to [K19]; thanks for these suggestions!

---

### Author Response · Authors · 2019-11-12
**Summary of revisions**

Dear Reviewers:

Thank you again for the valuable feedback! We have uploaded a revised version of the paper according to your suggestions. The main changes we made are:

[1] (AnonReviewer4) In both the algorithm description and experimental results sections, we added several sentences to make it clear that base optimizers can also use local momentum, and adding SlowMo on top of it can further improve the performance. We also added Table C.1 in the appendix which gives specific examples for $d^{(i)}_{t,k}$.

[2] (AnonReviewer3) We added two paragraphs in the related work section to discuss recent advances in understanding stochastic momentum methods and on fundamental limits for decentralized optimization.

[3] (AnonReviewer2) We rewrote the discussions about BMUF making clear that BMUF can be applied on top of different local optimizers such as SGD/Adam.  We emphasized that SlowMo can be built on local optimizers as well as decentralized methods.

We also thank all reviewers for the minor comments! We have clarified the discussion on Local SGD, changed “descent direction” to “update direction”, and revised the notation in algorithm 3.

---

### Public Comment · ~Albert_Zeyer1 · 2024-04-11
**Nesterov**

In the original BMUF, Nesterov momentum was also tested for the slow momentum, and this improved over the standard slow momentum. This is not addressed in the paper here, and it seems that only the standard slow momentum is used (although Nesterov momentum was used in the inner base optimizer in some experiments). I wonder whether there were any experiments on that.

---

### Decision · Program_Chairs · 2019-12-19

**Decision:**

Accept (Poster)

**Comment:**

This paper presents a new approach, SlowMo, to improve communication-efficient distribution training with SGD. The main method is based on the BMUF approach and relies on workers to periodically synchronize and perform a momentum update. This works well in practice as shown in the empirical results.

Reviewers had a couple of concerns regarding the significance of the contributions. After the rebuttal period some of their doubts were clarified. Even though they find that the solutions of the paper are an incremental extension of existing work, they believe this is a useful extension. For this reason, I recommend to accept this paper.